# Statistical and structural identifiability in representation learning

**Walter Nelson[1], Marco Fumero[1], Theofanis Karaletsos[2] & Francesco Locatello[1]**
[1]Institute of Science and Technology Austria
[2]Chan Zuckerberg Initiative

## Abstract

Representation learning models exhibit a surprising stability in their internal representations. Whereas most prior work treats this stability as a single property, we formalize it as two distinct concepts: **statistical identifiability** (consistency of representations across runs) and **structural identifiability** (alignment of representations with some unobserved ground truth). Recognizing that perfect pointwise identifiability is generally unrealistic for modern representation learning models, we propose new model-agnostic definitions of statistical and structural near-identifiability of representations up to some error tolerance $\epsilon$. Leveraging these definitions, we prove a statistical $\epsilon$-**near-identifiability** result for the representations of models with nonlinear decoders, generalizing existing identifiability theory beyond last-layer representations in e.g. generative pre-trained transformers (GPTs) to near-identifiability of the intermediate representations of a broad class of models including (masked) autoencoders (MAEs) and supervised learners. Although these weaker assumptions confer weaker identifiability, we show that independent components analysis (ICA) can resolve much of the remaining linear ambiguity for this class of models, and validate and measure our near-identifiability claims empirically. With additional assumptions on the data-generating process, statistical identifiability extends to structural identifiability, yielding a simple and practical recipe for disentanglement: ICA post-processing of latent representations. On synthetic benchmarks, this approach achieves state-of-the-art disentanglement using a vanilla autoencoder. With a foundation model-scale MAE for cell microscopy, it disentangles biological variation from technical batch effects, substantially improving downstream generalization.

## 1 Introduction

Despite the massive variety of data modalities, pretext tasks, training procedures, and datasets, disparate self-supervised learning models as a whole seem to be converging on a shared set of representations of the natural world (Huh et al., 2024) which are useful for a surprising variety of downstream tasks (Kraus et al., 2024; Hayes et al., 2025; Baevski et al., 2020; Brohan et al., 2023). A classical lens for studying this phenomenon is the notion of *identifiability* (Reizinger et al., 2025a). In likelihood-based statistical inference, identifiability is the condition that data are sufficient to completely characterize the parameters of the model (Casella & Berger, 2001). The situation is considerably trickier for neural network models: the parameter space is large and invariant to e.g. permutations of the neurons, and the training procedures might lack a likelihood-based interpretation. Instead, recent work in identifiability focuses on finding conditions such that infinite data is sufficient to characterize the trained model's *representations* of that data (Reizinger et al., 2025a) up to some equivalence class such as a linear transformation (Roeder et al., 2021).

We begin by sharpening existing definitions of representation identifiability, recognizing that existing results fall into two categories. The first is what we refer to as statistical identifiability, or the condition that optimizing a given representation learning model will yield the *same* representations up to some simple transformation. The second is what we refer to as structural identifiability, or the condition that optimizing a given representation learning model will yield a *particular* representation every time, corresponding to some latent component of the data-generating process. We provide

definitions of statistical and structural identifiability that relax the requirement that the representations are exactly identifiable, making them the first general-purpose formulations which are applicable to the case where the representations are "nearly" identifiable up to some error tolerance $\epsilon$ and extending prior model-specific cases (Nielsen et al., 2025; Buchholz & Schölkopf, 2024).

Leveraging these definitions, we prove several new identifiability results. Our first result shows that for models which have statistically identifiable outputs, such as generative pre-trained transformers, supervised classifiers, and encoder-decoder models (Roeder et al., 2021), the *intermediate-layer* representations are also statistically $\epsilon$-nearly identifiable up to a rigid transformation. Unlike several recent results, these representations can be mapped non-linearly to the loss, and $\epsilon$ is governed by a mild function class condition on the mapping from the intermediate layer to the identifiable outputs. Our second result shows that linear independent components analysis (ICA) can resolve this rigid indeterminacy, yielding near-identifiability up to signed permutations. Notably, our sharper definitions of statistical and structural identifiability reveal that these results are available without strong assumptions on the data-generating process, instead requiring only this mild function class assumption on the model, extending prior theoretical results on isometric and approximately isometric learning (Gresele et al., 2022; Buchholz & Schölkopf, 2024). Our final result shows that if one *is* willing to make a similar assumption on the data-generating process, encoder-decoder models which are statistically identifiable and achieve perfect reconstruction are also structurally identifiable. Notably, perfect reconstruction is another assumption which can be relaxed if statistical identifiability is all that is required.

In addition to our theoretical contributions, we perform a series of experiments to validate our claims. In synthetic experiments on autoencoders, we show that hyperparameter selection and regularization impacts the statistical identifiability nearness $\epsilon$ in ways that are predicted by our theory. Subsequent experiments show that near-identifiability also holds in off-the-shelf pre-trained models, and that the linear indeterminacies predicted by our theory can in practice be resolved by ICA. Next, we investigate whether our structural identifiability result can be applied to the special case of disentanglement (Locatello et al., 2020), finding that the simple combination of vanilla autoencoders and linear ICA applied to the latent space yields disentanglement on several benchmark datasets, competitive with some of the best existing models. Finally, we show that linear ICA applied to the latent space of a masked autoencoder for cell imaging successfully disentangles batch effects from biological variation, a key problem in the application of machine learning to biology.

## 2 RELATED WORK

**Statistical identifiability of representations**   Prior representation identifiability results make strong assumptions on the data-generating process (Zimmermann et al., 2021; Reizinger et al., 2025b; Khemakhem et al., 2020b; Chen et al., 2024; Lachapelle et al., 2023) or assume a linear relationship between the representations and the loss (Roeder et al., 2021; Marconato et al., 2025; Nielsen et al., 2025), and generally do not distinguish between statistical and structural identifiability as we do here. Our work directly addresses this gap by proposing a concrete definition of statistical $\epsilon$-near-identifiability which is provably met by the general-purpose models in widespread use today, with only mild assumptions on the model class and few assumptions on the data-generating process. We directly measure the consequences of our *statistical identifiability* results by assessing the $\ell_2$ convergence of representations in real-world models, extending prior work on representation similarity (Roeder et al., 2021; Huh et al., 2024; Klabunde et al., 2025; Nielsen et al., 2025; Marconato et al., 2025). Nielsen et al. (2025) relaxes the identifiability theory of Roeder et al. (2021) for generative pre-trained transformers, showing that the Kullback-Leibler divergence on the next-token distribution fails to serve as a witness for differences in the penultimate-layer representations, and proves a sufficient condition for divergences that do, which could be viewed as a particular form of near-identifiability for a particular model class. Reizinger et al. (2024) formalizes $\epsilon$-non-identifiability with respect to the KL divergence, showing that this failure generalizes beyond representational identifiability to other properties of interest for large language models. For a history of the term identifiability, see Appendix A.1.

**Structural identifiability of representations**   Ours is also the first work to make clear the distinction between such model-specific identifiability results and structural identifiability results such as disentanglement. We formalize an assumption on the data-generating process that allows us to

extend our key statistical identifiability result to structural identifiability of encoder-decoder models, and define and characterize a rich class of data-generating processes which meet this assumption. This result is similar to other works that aim to "invert the data-generating process" (Zimmermann et al., 2021; Reizinger et al., 2025b; Von Kügelgen et al., 2021), including theoretical work on the isometry assumption combined with ICA (Horan et al., 2021), some of which uses a different, average-case notion of near-isometry to claim near-recovery of the true latents in a nonlinear ICA model (Buchholz & Schölkopf, 2024). These prior works are far from real-world practice, with only Reizinger et al. (2025b) presenting any results on a real model with real data, showing that linear concept decoding is possible via parametric instance discrimination in ImageNet-X, but lacking a clear disentanglement result. In stark contrast, we illustrate a practical application of our theory by showing that we can improve out-of-distribution generalization in a real-world biological foundation model for cell microscopy via disentanglement. Our work also differs from prior work on causal representation learning (see Yao et al. (2025)) in that our structural identifiability result is completely unsupervised, relying on inductive biases rather than supervision in the form of interventions.

## 3    STABILITY THEORY

We begin by providing a novel theory of the stability of neural representations, with an eye toward self-supervised models. All theorems are presented informally with important constants in the main text, with full theorem statements, lemmata, proofs, and model-specific treatments in Appendix A.3.

### 3.1    STATISTICAL NEAR-IDENTIFIABILITY

We consider a data distribution supported on an arbitrary space $\mathcal{X}$. A representation learning model can be fully characterized by its parameter space $\Theta$, its loss function, and the deterministic mapping from parameters to representation functions. Concretely, for each $\theta \in \Theta$, let $\mathcal{L}_\theta : \mathcal{X} \to \mathbb{R}$ denote the corresponding loss function, and define the model as $\mathcal{M} = \{\mathcal{L}_\theta : \theta \in \Theta\}$. If some component of $\theta$ parameterizes a representation function $f_\theta : \mathcal{X} \to \mathbb{R}^D$, then identifiability theory in machine learning aims to characterize the properties of $f_\theta$ after training by minimizing $\mathbb{E}[\mathcal{L}_\theta(x)]$ yields a set of parameters $\theta$.

**Definition 1.** *Let $\mathbf{P}(x)$ denote some data distribution supported on $\mathcal{X}$. Consider a machine learning model $\mathcal{M} = \{\mathcal{L}_\theta : \theta \in \Theta\}$, and let $F : \theta \mapsto f_\theta$ be some deterministic transform of the model parameters yielding a representation function $f_\theta : \mathcal{X} \to \mathbb{R}^D$. Let $\mathcal{S} \subset \Theta$ be the set of minimizers of $\mathbb{E}_{x \sim \mathbf{P}(x)}[\mathcal{L}_\theta(x)]$. For some group $\mathcal{H}$ of functions from $\mathbb{R}^D$ to itself, we say that $(\mathbf{P}, \Theta, \mathcal{L}_\theta, F)$ is statistically $\epsilon$-nearly identifiable in the limit up to $\mathcal{H}$ if for every $\theta, \theta' \in \mathcal{S}$, we have $\|f_\theta - h \circ f_{\theta'}\| \leq \epsilon$ for some $h \in \mathcal{H}$.*

**Remark.** For the remainder of this paper, we will use the the $L^\infty$ norm (essential supremum with respect to $\mathbf{P}$) for functions taking values in the Euclidean space $\mathbb{R}^D$ endowed with the $\ell_2$ norm. When $\epsilon = 0$, we drop the $\epsilon$-nearly and refer to the triple simply as statistically identifiable in the limit up to $\mathcal{H}$. When the identifiability is pointwise, we refer to the model as statistically identifiable in the limit.

**Intuition.** This generalizes prior definitions of representation identifiability by the introduction of the "slack" term $\epsilon$. Definition 1 says that a model's representations as given by independent retrainings $f_\theta$ and $f_{\theta'}$ are near-identifiable if they are the same up to a simple transformation group $\mathcal{H}$ (e.g. rotations) and a small amount of distortion $\epsilon$. In this way, we also generalize the classical definition if identifiability from mathematical statistics (Casella & Berger, 2001), see Appendix A.1 for a history.

In this paper, we will mainly deal with near-identifiability of latent representations up to the function classes $\mathcal{H}_{\text{linear}}$, $\mathcal{H}_{\text{rigid}}$ and $\mathcal{H}_\sigma$. $\mathcal{H}_{\text{linear}}$ is the group of invertible linear transformations on $\mathbb{R}^D$, while $\mathcal{H}_{\text{rigid}}$ is the class of rigid linear transformations on $\mathbb{R}^D$, which consists of compositions of rotations, reflections and translations ($\mathcal{H}_{\text{rigid}} \subset \mathcal{H}_{\text{linear}}$). In practice, translations can be ignored by assuming e.g. zero mean of the representation distribution. Similarly, reflections only flip signs, and can usually also be ignored. It is therefore useful to imagine the main indeterminacy of $\mathcal{H}_{\text{rigid}}$ as being a special orthogonal matrix in $\mathbb{SO}(D)$, i.e. a rotation. $\mathcal{H}_\sigma$ is the class of signed permutations of $\mathbb{R}^D$, which are generally not resolvable because there is no reasonable way to specify an ordering to the latent

variables or signs to individual latent variables (should the $x$ coordinate of an object in a scene be represented left-to-right, or right-to-left?).

**Connection with prior results**   Existing representation identifiability results can easily be recast in our framework. For example, contrastive learning models using the InfoNCE loss with augmentation distributions satisfying a particular isotropy condition in latent space are identifiable up to $\mathcal{H}_{\mathrm{rigid}}$ (Zimmermann et al., 2021). This isotropy condition is impossible to validate in practice without access to the ground-truth data-generating factors, reflecting the fact that it is a strong assumption on the true data-generating process.

Another result is due to Roeder et al. (2021), which applies to models whose losses take the following form and can be interpreted as exponential family negative log-likelihoods:

$$\mathcal{L}_\theta(x, y) = -\boldsymbol{\eta}_\theta(x)^\mathsf{T}\mathbf{t}_\theta(y) + A_\theta(x) = -\log q_\theta(y \mid x) \tag{1}$$

where $q_\theta$ is the approximating distribution, $\mathbf{t}_\theta$ is a sufficient statistics function, $\boldsymbol{\eta}_\theta$ is the natural parameter function, and $A_\theta$ is the log partition function. As an example, $\mathbf{t}_\theta$ might map categorical labels to their corresponding vectors in a final linear weights matrix (covering the case of supervised multi-class classification, including next-token prediction). $\boldsymbol{\eta}_\theta$ maps inputs to their representations, which are shown to be identifiable up to $\mathcal{H}_{\mathrm{linear}}$ when pointwise equality of the losses is attained. In a short proof in Appendix A.3.1, we show that a simple sufficient richness condition on the approximating class extends this result to our definition of identifiable in the limit. Put together, this result means that "perfect optimization" on infinite data will yield a supervised or GPT-class model with the same penultimate-layer representations every time, up to some unknown linear transformation. An extension of this result provides for the same kind of identifiability on a linear subspace of the representation space, allowing for the case of models with representations of different ambient dimension (Marconato et al., 2025). Nielsen et al. (2025) provides for a further generalization, deriving a notion of distance on the space of modeled likelihoods $\mathcal{Q} = \{q_\theta(y \mid x) : \theta \in \Theta\}$ such that closeness in distribution implies closeness in the penultimate-layer representations given by $\boldsymbol{\eta}_\theta$, a form of near-identifiability (see also Appendix A.3.1).

Even these GPT results are limited because they only treat these penultimate-layer representations given by $\boldsymbol{\eta}$, which are mapped *linearly* to the loss. For many models, we're interested in representations from earlier layers which are mapped to the loss *nonlinearly*, such as with a nonlinear decoder or head. Our key result, captured in the following theorem, provides near-identifiability up to rigid transformations in such cases. The level of nearness $\epsilon$ is governed by the degree of local bi-Lipschitzness of this nonlinear mapping. This result allows us to treat earlier-layer representations in GPT or supervised classification models, for example, or the latent representations of (masked) autoencoders.

**Theorem 1.** *(Informal) Let $\mathbf{P}(x)$ be a data distribution, and let $\mathcal{M}$ be a model with a parameter space $\Theta$ and loss function $\mathcal{L}_\theta$. Let $F : \theta \mapsto f_\theta$, $G : \theta \mapsto g_\theta$ and $H : \theta \mapsto g_\theta \circ f_\theta$. Then, if $(\mathbf{P}, \Theta, \mathcal{L}_\theta, H)$ is statistically identifiable in the limit, then $(\mathbf{P}, \Theta, \mathcal{L}_\theta, F)$ is statistically $\epsilon$-nearly identifiable in the limit up to $\mathcal{H}_{\mathrm{rigid}}$ for $\epsilon = c_D\sqrt{2L + L^2}\Delta$ where $1 + L$ is a local bi-Lipschitz constant bound for $g_\theta$, and $c_D$ and $\Delta$ are constants independent of the model (and $L$).*

**Intuition.** Here, we give the first general-purpose identifiability result for the internal representations (i.e. arbitrary-layer) of a broad class of models, including (masked) autoencoders, next-token predictors, and supervised learners. $H$ parameterizes the end-to-end neural network $g_\theta \circ f_\theta$, which is assumed to have identifiable outputs: for example, when the loss is the mean squared error, the neural network learns the optimal function, namely the conditional mean. The identifiability of the internal representations given by $f_\theta$ (the "encoder") is then governed by the local bi-Lipschitzness of the function $g_\theta$ (the "decoder") mapping them to these identified outputs. The bi-Lipschitz constraint controls the degree to which $g_\theta$ deforms distances. Intuitively, a bound on the local bi-Lipschitz constant is small when small changes in the latent variables result in small changes in the outputs of the network. The proof is given in Appendix A.3.2, and we provide concrete examples for a number of architectures, including masked autoencoders, supervised learners, and GPTs in Appendices A.3.3 and A.3.4.

This result is the most general we are aware of for quantifying representation identifiability. The local bi-Lipschitz condition is difficult to test empirically, but prior work has shown that many popular regularization techniques push neural networks toward a state of "dynamical isometry", which can be

viewed as a bi-Lipschitz condition (Xiao et al., 2018; Bachlechner et al., 2020; Miyato et al., 2018; Karras et al., 2020; Zhang et al., 2019) due to the singular values of the Jacobian concentrating near one. We derive the precise relationship in Appendix A.4.

## 3.2 RESOLVING LINEAR INDETERMINACIES WITH ICA

Later, we will illustrate applications of the near-identifiability result Theorem 1 to both vanilla autoencoders and masked autoencoders. To do this, we will find it useful to resolve the remaining linear indeterminacy in the latent space posed by $\mathcal{H}_{\text{linear}}$ or $\mathcal{H}_{\text{rigid}}$. We propose to do this by applying independent components analysis to the latent representations. We do not provide any novel ICA identifiability results in this work. Rather, we show that our conception of $\epsilon$-nearness in identifiability poses no further complications for the downstream application of ICA. This is partly a generalization of an earlier result by Horan et al. (2021), which covers the perfectly identifiable case and first proposed combining isometric learning (in the form of the Hessian locally linear embedding algorithm, Donoho & Grimes (2003)) with ICA. The statement of our theorem (and its corollary Theorem 3) is similar to Theorem 3.1 in Buchholz & Schölkopf (2024) for nonlinear ICA, which relaxes the required pointwise constraint on the Jacobian implied by bi-Lipschitzness in exchange for an $L_2$ identifiability bound rather than $L_\infty$.

**Theorem 2.** *(Informal) Suppose $(\mathbf{P}, \Theta, \mathcal{L}_\theta, F)$ is statistically $\epsilon$-nearly identifiable up to $\mathcal{H}_{\text{linear}}$ for $F : \theta \mapsto f_\theta$. Then, for a new model with parameter space $\Theta'$ and loss $\mathcal{L}'_\theta$ which applies whitening and contrast function-based independent components analysis to the latent representations given by $f_\theta$, and $F' : \theta \mapsto f'_\theta$ which yields the transformed representations, $(\mathbf{P}, \Theta', \mathcal{L}'_\theta, F')$ is statistically $\epsilon'$-near-identifiable up to $\mathcal{H}_\sigma$ for $\epsilon' = K\epsilon + K'\epsilon^2$, where $K$ and $K'$ are constants free of $\epsilon$ that depend on the spectrum of the covariance matrix of the representations and the properties of the ICA contrast function.*

**Intuition.** Consider some representations in a model which are linearly identifiable, such as the penultimate layer of a GPT-class model or the latent tokens of a masked autoencoder covered by Theorem 1. Whitening reduces the linear indeterminacy to a rigid one, while ICA (if sufficiently well-converged) resolves the final rigid indeterminacy to a signed permutation, with nearness preserved (up to new constants) along each step.

## 3.3 FROM STATISTICAL TO STRUCTURAL IDENTIFIABILITY

While statistical identifiability on its own may be a useful property for reliability and analysis, there has been recent interest in structural identifiability of representations, or the ability of the model to recover some latent component of the data-generating process which is useful for some downstream tasks. Below, we formalize the distinction between the two.

**Definition 2.** *Let $\mathbf{P}(u)$ denote a distribution over some unobservable parameters with support $\mathcal{U} \subseteq \mathbb{R}^D$. Let $\mathbf{P}(x \mid u)$ denote some conditional distribution such that $u(x) = \arg\sup_{u \in \mathcal{U}} \mathbf{P}(x \mid u)$ is well-defined almost everywhere with respect to $\mathbf{P}(x)$, where $\mathbf{P}(x) = \int \mathbf{P}(x \mid u)\mathbf{P}(u)\,du$ is the marginal distribution of the data with support $\mathcal{X}$. Consider a machine learning model $\mathcal{M} = \{\mathcal{L}_\theta : \theta \in \Theta\}$, with solutions $\mathcal{S} \subset \Theta$ to the minimization of $\mathbb{E}_{x \sim \mathbf{P}(x)}[\mathcal{L}_\theta(x)]$. Let $F : \theta \mapsto f_\theta$ be a deterministic transform of the parameters yielding a representation function $f_\theta : \mathcal{X} \to \mathbb{R}^D$. For some group of functions $\mathcal{H}$ from $\mathbb{R}^D$ to itself, we say that $(\mathbf{P}, \Theta, \mathcal{L}_\theta, F)$ $\epsilon$-nearly identifies the structure $u$ up to $\mathcal{H}$ if for all $\theta \in \mathcal{S}$, we have that $f_\theta$ satisfies $\|h \circ f_\theta - u\| \leq \epsilon$ for some $h \in \mathcal{H}$.*

**Intuition.** Statistical identifiability is in some sense *weaker* than structural identifiability. Statistical identifiability is the condition that the representations are consistent, while structural identifiability is the condition that the representations are consistently "correct". In order to define utility, or correctness, we need to assume the existence of some latent component of the data-generating process that we're aiming to recover, $u$. The well-studied setting of disentanglement (Locatello et al., 2019) represents a special case where $\mathbf{P}(u)$ is assumed to have independent components. As an example, in Section 4.4, we consider the situation where $\mathbf{P}(u)$ is a distribution over natural latent biological factors along with independent technical variates, and $x$ is generated via a smooth function of $u$ with smooth inverse, leaving $u(x)$ well-defined. In Appendix A.5, we provide a simple proof that structural identifiability implies structural identifiability, including in the $\epsilon$-near case provided that $\mathcal{H}$ is bounded in the sense of an operator norm.

Finally, we make precise the assumptions on the data-generating process necessary to extend our statistical identifiability result in Theorem 2 to structural identifiability. We show that bi-Lipschitz data-generating processes are structurally identified by reconstructing encoder-decoder models which are nearly identifiable up to $\mathcal{H}_{\text{rigid}}$ in the sense of Theorem 1, and up to $\mathcal{H}_\sigma$ when combined with ICA as in Theorem 2. Because in this setting the assumption made on the data-generating process (bi-Lipschitzness) is the same as on the model, structural identifiability is a fairly straightforward corollary of the statistical identifiability in Theorem 2. Notably, however, Theorem 3 requires perfect reconstruction in an autoencoding-type model, while Theorem 2 only requires identifiable outputs, and is therefore significantly more general.

**Theorem 3.** *(Informal) Let $\mathbf{P}(u)$ be some multivariate non-Gaussian distribution with independent components and consider data $\mathbf{P}(x)$ generated by pushforward through a smooth diffeomorphism $g$ such that $g$ is $(1 + \delta)$-bi-Lipschitz. Let $\mathcal{M}$ be a model with a sufficiently rich parameter space $\Theta$. Let $F : \theta \mapsto f_\theta$, $G : \theta \mapsto g_\theta$ and $H : \theta \mapsto g_\theta \circ f_\theta$. Then, if $(\mathbf{P}, \Theta, \mathcal{L}_\theta, H)$ structurally identifies the identity function in the limit (i.e. attains perfect reconstruction), we have that $(\mathbf{P}, \Theta, \mathcal{L}_\theta, F)$ $\epsilon$-nearly identifies the structure $g^{-1}$ up to $\mathcal{H}_{\text{rigid}}$, and furthermore that a new model $\mathcal{M}'$ which applies whitening and independent components analysis to the latent representations given by $f_\theta$ $\epsilon'$-nearly identifies the structure $g^{-1}$ up to $\mathcal{H}_\sigma$ where $\epsilon$ and $\epsilon'$ depend on $\delta$ and Lipschitz bounds on $g_\theta$, and $\epsilon'$ depends additionally on the spectrum of the covariance matrix of the representations and the properties of the ICA contrast function employed.*

**Intuition.** Structural identifiability is *stronger* than identifiability, so we require additional assumptions on the data-generating process to achieve it. In particular, here we assume that the data-generating function mapping "true" latents to observables is locally bi-Lipschitz, which combined with independence and non-Gaussianity is sufficient to nearly recover the true latents via ICA for any nearly identifiable reconstructing model with a locally bi-Lipschitz decoder. The proof is given in Appendix A.3.6.

### 3.3.1 BI-LIPSCHITZ DATA-GENERATING PROCESSES

Naturally, it's useful to characterize what kinds of data-generating processes might be covered by Theorem 3. Several interesting image data-generating processes are known to approximately satisfy a Euclidean isometry condition (which is equivalent to a local 1-bi-Lipschitz constraint for smooth mappings) such as smooth articulations of cartoon faces (Tenenbaum et al., 2000; Horan et al., 2021). Furthermore, the success of regularization techniques similar to isometry constraints in diverse classes of neural network models in real-world settings suggests it is a useful inductive bias in practice as well (Karras et al., 2020; Lee et al., 2022). In the rest of this section, we aim to better characterize what these assumptions mean. Specifically, we give some examples of nearly isometric data-generating processes inspired by the popular `dSprites` dataset (Matthey et al., 2017) and show that disentanglement in this setting implies the structural identification of the true data-generating factors, using a technique developed by Grimes (2003).

**Example** Consider a continuous relaxation of images, where a square black-and-white image is represented by an $L^2$ function $\iota : [-1, 1] \times [-1, 1] \to \{0, 1\}$, with $\iota(x, y)$ giving the value of the $(x, y)$th "pixel". As an example, the image of a white square with radius $0 < r < 1$ in the centre of the "frame" is given by $\iota(x, y) = \mathbb{I}[|x| \le r, |y| \le r]$ where $\mathbb{I}$ is the indicator function. One can first imagine a manifold of such images where the centre $p \in [a, b]$ of the square is moved from left to right. We write this as a continuum of images produced by the smooth function $f : [a, b] \to L^2$ where $a + 1 \ge r$ and $1 - b \ge r$ to ensure that the square does not leave the frame. In this case, each "image" on the continuum $f(p)$ is the function $(x, y) \mapsto \mathbb{I}[|x - p| \le r, |y| \le r]$, where $p$ is the square's $x$ coordinate.

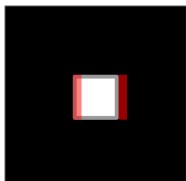

Figure 1: A simple isometric data-generating process.

We'll show that $f$ is a local isometry, meaning that it preserves a notion of distance perfectly. This is equivalent to a 1-bi-Lipschitz constraint. To see this, consider the Gateaux derivative of $f$, which is

just the limiting behaviour of articulating the square $\epsilon$ units to the right:

$$f'(p) = \lim_{\epsilon \to 0} \frac{f(p+\epsilon) - f(p)}{\epsilon}$$

$$= \lim_{\epsilon \to 0} \frac{1}{\epsilon} \left[ \underbrace{\mathbb{I}[p+r < x < p+r+\epsilon, |y| < r]}_{\text{``gained'' white pixels}} - \underbrace{\mathbb{I}[p-r-\epsilon < x < p-r, |y| < r]}_{\text{``lost'' white pixels}} \right]$$

where the indicators in the limit represent the pixels that change when articulating the square $\epsilon$ units to the right. Intuitively, shifting the square $\epsilon$ units to the right only changes $\epsilon r$ pixels to the right (flips them from black to white) and $\epsilon r$ pixels to the left (flips them from white to black) of the original square. The situation is drawn in Figure 1, where the white square has a gray border for clarity and the red shaded areas show the white pixels which are gained and lost by shifting the square $\epsilon$ units to the right. The isometry condition considers the situation as $\epsilon \to 0$ (intuitively, as the width of the red rectangles shrinks to zero).

By an argument made formal in Appendix A.6, we can rely on something like preservation of the $L^2$ norm under limits to have $||f'(p)||^2 = 2r$, which is constant when $r$ is fixed. In the univariate case, this is sufficient for $f$ to locally preserve a notion of distance. In particular, for any two values of the latent $p_0$ and $p_1$ we have

$$|p_1 - p_0| \propto \int_{p_0}^{p_1} ||f'(p)||^2 \, dp = 2r|p_1 - p_0|$$

where the integral is the usual geodesic distance along the manifold of "images". Due to the constant $2r$, some literature refers to these as scaled isometries (Lee et al., 2022). In practice, we can typically ignore the scaling constant.

While sprites datasets are known to be overly simplistic, this analysis provides some intuition that interesting real-world image manifolds might usefully be approximated by isometries, or functions that are *nearly* isometries. For example, a similar analysis we defer to Appendix A.6 is illuminating for the multivariate case where both the radius $r$ of the square and its $x$-coordinate $p$ are varied. In this setting, we have that the Gateaux derivative is non-constant in $r$ (specifically, the map is conformal). However, if we assume a compact support for the latents, i.e. that $p \in [a, b]$ and $r \in (0, R]$, the data-generating process is additionally $B$-bi-Lipschitz with the constant $B$ dependent on $a$ and $b$, and Theorem 3 applies.

## 4 EXPERIMENTS

We perform four sets of experiments: direct validation of Theorem 1 on MNIST using vanilla autoencoders (Section 4.1), direct validation of Theorems 1 and 2 in off-the-shelf pretrained self-supervised learning models (Section 4.2), an application of Theorem 3 to a classic disentanglement problem in several synthetic datasets (Section 4.3), and a real-world application to deconfounding for out-of-distribution generalization in a real-world foundation model for cell microscopy in biology (Section 4.4).

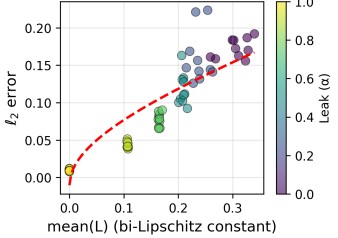

Figure 2: Controlling the bi-Lipschitz constant $L$ leads to improved identifiability (reduced $\ell_2$ error).

### 4.1 WARMUP: CONTROLLING IDENTIFIABILITY

We begin with experiments in a regime where the local bi-Lipschitz constant can be controlled as directly as possible, and examine whether our theory correctly predicts the level $\epsilon$ of near-identifiability. We consider fully-connected autoencoders with 3-layer encoders and decoders, and orthogonal linear layers with LeakyReLU activations (with leak parameter $\alpha \in [0, 1]$). The local bi-Lipschitz constant of the decoders is therefore bounded by $1/\alpha^K$ where $K = 3$ is the number of layers in the decoder. For $\alpha = 1$, the network is linear, while for $\alpha = 0$, it's a ReLU network. We fit pairs of autoencoders with different initializations

and seeds to MNIST (LeCun et al., 2010), and assess the relationship between reconstruction error, empirical near-identifiability, and empirical measurements of the local bi-Lipschitz constant, which is manipulated by varying $\alpha$ between 0 and 1. According to the proportionality in Theorem 1, we estimate how well the empirically estimated bi-Lipschitz term $\sqrt{L + L^2}$ predicts identifiability, as measured by the average $\ell_2$ error from the optimal rigid transformation between the pair of latent spaces. Results are summarized in Figure 2. Full experimental details are available in Appendix A.7.1.

## 4.2 MEASURING IDENTIFIABILITY OF PRE-TRAINED MODELS

Our next aim is to validate the statistical near-identifiability up to linear (for GPT-class models, Theorem 1 of Roeder et al. (2021)) and rigid (for autoencoder-class and supervised models, Theorem 1) transformations predicted by theory, and the ability of ICA to resolve the remaining linear indeterminacy (Theorem 2).

Matching our theory, we examine pairs of models that have the same architecture, loss, and are trained on the same dataset independently. Rigid similarities (rigid transforms with a scaling constant to allow for varying regularization across model pairs), linear transformations and ICA transforms are estimated between representation spaces. We measure near-identifiability with the average $\ell_2$ error in the self-supervised model's representation space, along with the efficiency of the ICA transform as the percentage reduction of $\ell_2$ error relative to the rigid transform (since the degrees of freedom are roughly the same). Results are shown in Table 1.

GPT-class models exhibit excellent linear alignment, as predicted by the theory of Roeder et al. (2021). As predicted by our theory, MAEs exhibit rigid alignment up to a similar level of error, notably including one example across model sizes. In all cases, ICA mitigates a substantial portion of the indeterminacy due to the linear variation, notably without any supervision. In particular, for MAE models, ICA is nearly 60% as efficient as computing the optimal rigid transform between the two models in a fully-supervised fashion. Full experimental details are available in Appendix A.7.2.

| Model Pair | Permutation | Supervised | | ICA (% eff.) |
|---|---|---|---|---|
| | | Rigid | Linear | |
| Pythia-160M-0 $\rightarrow$ Pythia-160M-1 | 0.219 | 0.150 | 0.131 | 0.202 (25%) |
| MAE-`timm` $\rightarrow$ MAE-original | 0.197 | 0.109 | 0.036 | 0.145 (59%) |
| CheXpert-small $\rightarrow$ CheXpert-base | 0.218 | 0.104 | 0.048 | 0.175 (38%) |
| ResNet-18-fc-1 $\rightarrow$ ResNet-18-fc-2 | 0.382 | 0.206 | 0.175 | 0.312 (40%) |

Table 1: Supervised and unsupervised alignment scores between pairs of models which measure empirical identifiability. The optimal transforms (permutation, rigid, linear, or ICA) are estimated between the two models, and average $\ell_2$ errors normalized by latent diameter are reported. For ICA, we also report its efficiency as the reduction of $\ell_2$ error from the permutation to the rigid transform.

## 4.3 DISENTANGLEMENT USING VANILLA AUTOENCODERS

Next, we assess whether our theory correctly predicts structural identifiability of the ground-truth data-generating factors in synthetic datasets matching the assumptions of Theorem 3. We examine vanilla autoencoders, an appealing architecture due to their simplicity. In such simple models, weight decay is known to be sufficient to regularize the Lipschitz constant of the decoder, thus making it a good testbed for our theory (Zhang et al., 2019).

Table 2: Disentanglement metrics (*InfoM*, *InfoE*, *InfoC*), of which *InfoM* and *InfoE* are the most important. AE + ICA performs comparably to some of the best disentanglement-specific neural networks, with almost no tuning. Results marked with (*) are quoted without reproduction from Hsu et al. (2023). Bolded metrics have the highest point estimates. Full details in Appendix A.7.3.

| model | aggregated | Shapes3D | MPI3D | Falcor3D | Isaac3D |
|---|---|---|---|---|---|
| | | (*InfoM* | *InfoE* | *InfoC*) $\uparrow$ | |
| AE | (0.39 0.76 0.25) | (0.34 **0.99** 0.16) | (0.42 0.40 0.31) | (0.37 **0.83** 0.20) | (0.41 0.80 0.34) |
| $\beta$-VAE* | (0.59 0.81 0.55) | (0.59 **0.99** 0.49) | (0.45 **0.71** 0.51) | (**0.71** 0.73 0.70) | (0.60 0.80 **0.51**) |
| $\beta$-TCVAE* | (0.58 0.72 **0.59**) | (0.61 0.82 **0.62**) | (**0.51** 0.60 **0.57**) | (0.66 0.74 **0.71**) | (0.54 0.70 0.46) |
| BioAE* | (0.54 0.75 0.36) | (0.56 0.98 0.44) | (0.45 0.66 0.36) | (0.54 0.73 0.31) | (0.63 0.65 0.33) |
| AE + ICA (ours) | (**0.65 0.83** 0.40) | (**0.79 0.99** 0.52) | (0.44 0.66 0.31) | (**0.71 0.83** 0.33) | (**0.64 0.82** 0.43) |

We use a well-established experimental testbed for assessing unsupervised disentanglement, specifically following the exact experimental protocol from Hsu et al. (2023). As baselines, we include a $\beta$-VAE (Higgins et al., 2017), $\beta$-total correlation VAE (Chen et al., 2018), and BioAE (Whittington et al., 2023), all of which leverage specialized regularization to achieve disentanglement. For comparison, we follow the supervised model selection strategy of Hsu et al. (2023), which shows best-case performance (Locatello et al., 2020).

For each dataset, we train a vanilla autoencoder with the only hyperparameter we vary being weight decay. Then, ICA is applied to its latent space. Models are evaluated on four datasets. Shapes3D is a toyish sprites dataset (Hsu et al., 2023; Burgess & Kim, 2018). Falcor3D and Isaac3D consist of rendered images of a living room and kitchen, respectively (Nie et al., 2020). MPI3D consists of real images of a real-world robotics setup (Gondal et al., 2019). Disentanglement of the learned latents is evaluated according to InfoMEC (Hsu et al., 2023) which consists of three complementary metrics: modularity (the degree to which each learned latent encodes only one true source), explicitness (the degree to which the latents capture all information about a source), and less important, compactness (the degree to which each source is encoded in only one latent). InfoMEC aims to resolve many of the issues with the DCI framework of disentanglement evaluations, including removing the need to select hyperparameters which can affect results (Hsu et al., 2023).

Vanilla autoencoders with ICA in latent space outperform specialized disentanglement models most of the time (Table 2), and on average perform better than all. Experimental runtime is roughly 6 hours per autoencoder hyperparameter setting on a single GPU (roughly 720 GPU hours total).

## 4.4 Deconfounding at foundation model-scale

High-throughput screens have become a critical tool in modern biology, particularly for drug discovery (Chandrasekaran et al., 2023). One example of such screenings is cell painting (Chandrasekaran et al., 2023; Sypetkowski et al., 2023), where a perturbation is applied (or not) to a collection of cells, which are then stained and imaged. A key challenge in the application of machine learning to these data is the presence of complex technical variation ("batch effects") that is not biologically significant (Arevalo et al., 2024; Chandrasekaran et al., 2023; Lin & Lu, 2022; Sypetkowski et al., 2023; Ando et al., 2017). For example, data collected from different microscopes, labs, or even just in different experiments can exhibit variation that is not of interest to the practitioner, potentially confounding results. A fundamental task in this setting is to quantify the degree to which a perturbation has a significant effect, which is challenging given that we have access only to high-dimensional, noisy observations in the form of images confounded by batch effects (Bereket & Karaletsos, 2023). In particular, we almost never have the ability to measure all sources of batch variation, and so it is of specific interest to be able to disentangle technical from biological variation without supervision.

We explore an application of Theorem 3 by applying independent component analysis to the latent space of OpenPhenom (Kraus et al., 2024), a large, open masked autoencoder trained on Rxrx3-core, a large library of cell painting images (Kraus et al., 2025). Downstream, we consider the task where the inferred embeddings are used to predict whether a given perturbation has been applied (i.e. classify "control" vs. "perturbed"). Because we are primarily interested in whether the procedure enhances

Table 3: Results from a downstream perturbation classification task. Each row represents a gene consisting of (#) separate experiments with different CRISPR guides. All experiments use the same set of 22,062 controls. Base = untransformed embeddings, PCA = whitened embeddings, PCA + ICA = whitened embeddings with ICA rotation applied, PCA + Rand = whitened embeddings with a random rotation applied.

| Gene | Mean AUROC (↑) | | | | Sparsity (↑ more sparse) | | | |
|---|---|---|---|---|---|---|---|---|
| | Base | PCA | PCA + ICA | PCA + Rand | Base | PCA | PCA + ICA | PCA + Rand |
| CYP11B1 (1) | 0.663 | 0.692 | **0.709** | 0.678 | 0.184 | 0.204 | **0.237** | 0.188 |
| EIF3H (1) | 0.682 | 0.724 | **0.749** | 0.725 | 0.192 | 0.224 | **0.268** | 0.214 |
| HCK (1) | 0.670 | 0.693 | **0.711** | 0.668 | 0.156 | 0.208 | **0.241** | 0.166 |
| MTOR (6) | 0.663 | 0.690 | **0.705** | 0.679 | 0.166 | 0.201 | **0.233** | 0.186 |
| PLK1 (6) | 0.803 | 0.811 | **0.815** | 0.792 | 0.251 | **0.307** | 0.305 | 0.262 |
| SRC (1) | 0.660 | 0.694 | **0.706** | 0.676 | 0.170 | 0.214 | **0.240** | 0.184 |

out-of-distribution generalization, we consider downstream classifiers trained on a subset of batches and evaluated on a held-out subset of batches.

**Inference & evaluation**    We perform inference on all images from Rxrx3-core and estimate the whitening and independent components analysis models at the patch level by randomly subsampling a patch from each image. We consider four conditions: the original embedding (Base), the embedding whitened with principal components analysis (PCA), the whitened embedding rotated using ICA (PCA + ICA), and as a baseline, the whitened embedding rotated randomly (PCA + Rand). Plates are used as the batch indicator, while a single patch-level embedding is used for each image. For each embedding condition (raw, whitened, whitened + ICA, whitened + random rotation), we hold out 20% of plates and train a gradient boosting classifier (Ke et al., 2017) on the remaining 80% of plates in a $k$-fold cross-validation scheme. Because some plates do not contain any perturbed samples (i.e. are entirely controls), we ensure that this split is roughly stratified on the label ("perturbed" vs. "control"). Perturbation classification is evaluated by the area under the receiver operator characteristic curve (AUROC). Embedding inference took approximately 1 hour on a single GPU, while ICA estimation took about 1 hour on 128 CPU cores with 128 GB RAM.

**Results**    Whitening alone often enhances the performance of downstream classification from the embeddings. The application of ICA consistently improves the performance even further (Table 3). To understand the source of the improvement, we measure sparsity with a measure called Hoyer sparsity which characterizes how biased trees are toward selecting a particular subset of features (Hoyer, 2004) defined as $\text{Sparsity}[\mathbf{c}] = \frac{\sqrt{D} - \frac{1}{\|\mathbf{c}\|_2}}{\sqrt{D}-1}$ for a $D$-dimensional vector of split fractions $\mathbf{c}$ such that $\|\mathbf{c}\|_1 = 1$ and where $c_d$ measures the fraction of times the $d$th variable was used in a split (higher $\rightarrow$ more important). The sparsity score is zero when all features are used equally often. Sparsity increases markedly with both whitening and ICA.

However, this measure does not specifically show that the information being ignored as a result of the increased sparsity specifically has to do with the distinction between technical and biological variation. To assess this, we measure how well the information useful for predicting the biological effect is concentrated in the top $k\%$ of predictors. Denote by $\mathbf{z}_{k\%}$ the top $k\%$ most important features for the prediction of the biological effect $y$ of interest, and by $\mathbf{z}'_{k\%}$ the remaining features. Then, the concentration is given by $\text{Concentration}[y] = \frac{\text{AUROC}[y;\mathbf{z}_{k\%}]}{\text{AUROC}[y;\mathbf{z}'_{k\%}]} - 1$, i.e., the improvement in predicting perturbation $y$ from the top features versus the bottom.

| Model | Concentration ($\uparrow$) |
|---|---|
| Base | 0.163 |
| PCA | 0.332 |
| PCA + ICA | **0.386** |
| PCA + Rand | 0.287 |

Table 4: Concentration of biological variation in the top 25% of features.

The concentration increases uniformly with whitening and with ICA (Table 4.4), even in the case of PLK1 guides where it does not confer a substantial gain in out-of-distribution AUROC. The results are not sensitive to reasonable values of $k$ (Appendix A.7.4). Interestingly, the results suggest that whitening alone biases the representation toward becoming axis-aligned even without ICA.

## 5    DISCUSSION

We have developed a theory of statistical near-identifiability of neural representations which is applicable to the internal representations of real-world self-supervised models. Notably, in contrast to prior work, our result requires few assumptions on the data-generating process, instead trading these off for assumptions on the model alone, and applies to a broad class of models including supervised learners, next-token predictors, and self-supervised learners. Additionally, we have shown that additional assumptions on the data-generating process can confer an even stronger result: namely, provable structural identifiability of the latent variables which generated the observables. We directly test our theory in real-world, off-the-shelf, pretrained self-supervised models. Furthermore, we leverage our theory to motivate the application of ICA to the latent spaces of self-supervised models and show that it can achieve state-of-the-art disentanglement results, including some of the first disentanglement results for out-of-distribution generalization in real-world data.

**Limitations & future work**    The primary limitation of our work is the difficulty in empirically testing the bi-Lipschitz assumptions necessary for our theory. Instead, we test the downstream

effects of our theory in four sets of experiments, and offer arguments from prior work which show that common regularization techniques which enable training of practical-scale neural networks (often referred to as "dynamical isometry", see also Appendix A.4) may lead to this condition. Interestingly, because the local bi-Lipschitz assumption is largely agnostic to data modality and model implementation details, it potentially applies to a broad class of both data-generating processes and models. As such, it may be an interesting lens for studying the phenomenon of cross-model representation convergence (Maiorca et al., 2023; Fumero et al., 2024), which is largely unaddressed by existing theory because existing identifiability results each require different assumptions on the data-generating process for different models (Huh et al., 2024; Reizinger et al., 2025a). Although we echo the calls of Reizinger et al. (2025a) for extensions to the practical regime (e.g. finite samples, imperfect optimization), we do not treat this case here, although it could be an extension of our framework. Additionally, because ours are the first identifiability results that apply to the intermediate layers of transformer-based next-token predictors, they may be useful for the interpretation of these models (Basile et al., 2025). In particular, Liu et al. (2025) show that discrete concept models learned atop last-layer GPT representations render the entire model end-to-end linearly identifiable, and the results of our paper suggest that this technique may work for intermediate-layer representations as well.

### ACKNOWLEDGMENTS

This work was supported by the Chan Zuckerberg Initiative (CZI) through the AI Residency Program. We thank CZI for the opportunity to participate in this program and the CZI AI Infrastructure Team for support with the GPU cluster used to train our models.

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

## A  APPENDIX

### A.1  A HISTORY OF THE TERM "IDENTIFIABILITY"

**Statistical identifiability**   Identifiability has a long history in statistics. For example, consider the following definition from Casella & Berger (2001) [p. 548], a canonical textbook in mathematical statistics:

> **Definition.**  A parameter $\theta$ for a family of distributions $\{f(x \mid \theta) : \theta \in \Theta\}$ is identifiable if distinct values of correspond to distinct pdfs or pmfs. That is, if $\theta \neq \theta'$, then $f(x \mid \theta)$ is not the same function as $f(x \mid \theta')$.

Our Definition 1 in the main text makes a straightforward generalization from likelihoods $f$ to losses $\mathcal{L}$. Further, because of the dominance of empirical risk minimization and the fact that, unlike most likelihoods in statistical inference, losses are non-convex, we define identifiability at the minimizers of the expected loss. In other words, our definition of identifiability agrees with the statistical one, modulo some small changes to make it useful for talking about non-convex optimization, specifically the empirical minimization of non-convex risk functions. Indeed, this means that not only does Definition 1 agree with the statistical definition, but also other recent attempts at defining and proving representation identifiability results for practical machine learning models (Roeder et al., 2021; Reizinger et al., 2024).

**Structural identifiability**   On the other hand, there is a similarly long history of "structural identifiability" from econometrics. For example, consider Koopmans & Reiersøl (1950), which partly led to the Nobel prize:

> **Identifiability of structural characteristics by a model.** It is therefore a question of great practical importance whether a statement converse to the one just made is valid: can the distribution $H$ of apparent variables, generated by a given structure $S$ contained in a model $\mathcal{S}$, be generated by only one structure in that model?

Clearly, this is a different definition. Indeed, Koopmans & Reiersøl acknowledge that it is different from the statistical definition above. It implicitly assumes the existence of a "true model" $S$ living within the estimable model class $\mathcal{S}$ (and that this model generated the data), which flies in the face of mathematical statistics' common quip "all models are wrong."

Pearl (1995) inherits this definition, perhaps contributing to its later infusion into the ICA literature:

> **DEFINITION 4 (Identifiability).**  The causal effect of $X$ on $Y$ is said to be identifiable if the quantity $P(Y|X)$ can be computed uniquely from any positive distribution of the observed variables that is compatible with a graph $G$.

Put simply, both these definitions of structural identifiability assume that the true data-generating process matches the model, and under this assumption, consider whether some structural parameter of interest can be identified. As a consequence, we refer to our Definition 2 as "structural identifiability". The only generalization we make is that we do not specifically enumerate $\mathcal{S}$, or indeed require the true structure $S \in \mathcal{S}$. We do not enumerate $\mathcal{S}$ because instead we enumerate $\Theta$, the space of possible parameters of the neural network model, and specify the mapping $F$ which generates the learned structure from a setting of the parameters $F : \theta \mapsto f_\theta$. We refer to the "true structure" as $u$. The relaxation of the requirement that the model class contains the "true structure" allows for other interesting cases, as illustrated in Example 2 below.

**Identifiability in linear ICA**  Occasionally, these definitions of identifiability are used somewhat interchangeably. This is particular true in the case of recent developments in independent components analysis, such as extensions to the nonlinear mixing regime. Interestingly, the main result of the original seminal linear ICA paper (Comon, 1994) is a statistical identifiability result, not structural (we edit the quoted Corollary slightly so that it is self-contained):

> **Corollary 13.** Let no noise be present in the linear ICA model with observations $\mathbf{y}$, and define $\mathbf{y} = M\mathbf{x}$ and $\mathbf{y} = F\mathbf{z}$ for a random variable $\mathbf{x}$ with independent components such that at most one is Gaussian. Then if $\Psi$ (a contrast function taking densities as inputs) is discriminant, $\Psi(p_\mathbf{x}) = \Psi(p_\mathbf{z})$ if and only if $F = M\Lambda P$ where $\Lambda$ is an invertible diagonal matrix and $P$ a permutation.

Of note, linear ICA is a case where most existing identifiability results depend on well-specification of the model. In particular, if the data is not generated by linear mixing from independent components, there is no guarantee that a linear decomposition into independent components exists (Castella et al., 2013) and therefore the statistical identifiability claim above is vacuous. In other words, there are few or no results available for statistical identifiability of linear ICA that don't also imply structural identifiability. This perhaps helps explain why the two concepts have been used somewhat interchangeably in this area of the literature. However, as we will see in the examples below, this is not necessarily the case for all models of interest.

Notably, LiNGAM, a classical approach which extends linear ICA to causal discovery Shimizu et al. (2006), provides an early example of this distinction between statistical and structural identifiability extending to graphical structures. In particular, they show that the independent components estimate can uniquely recover a causal graph under the assumption of linear functional relationships and additive noise (statistical identifiability). Structural identifiability follows under the assumption that a LiNGAM generated the data.

**Identifiability in non-linear ICA**  Noting that in general, non-linear extensions of ICA are impossible (Locatello et al., 2019), recent works have attempted to find sets of assumptions that render the task tractable. One line of work utilizes side information, showing that independence conditional on this side information leads to identifiability (Khemakhem et al., 2020a). Notably, Khemakhem et al. (2020a) explicitly define their identifiability as statistical identifiability (up to an equivalence class), but point out that if the true data-generating process takes the same form, structural identifiability of the true latents is achieved.

Below, we show how two other examples fit into these definitions.

**Example 1: mis-specified linear regression**  The first example comes from traditional statistics. Consider the identifiability of the usual linear model $y = x^\intercal \beta + \eta'$ when the model is mis-specified in the sense that the true data-generating process is $y = f(x) + \eta'$ for some non-linear $f$. If $f$ meets certain conditions, the following facts are true:

1. $\beta$ is statistically identifiable under the usual OLS assumptions (i.e. identifiable according to our Definition 1),

2. $f$ is not structurally identifiable (i.e., $\epsilon$-nearly structurally identifiable with $\epsilon = 0$) according to Definition 2,

3. the model is not absolutely identifiable according to Reizinger et al. (2025a), because the data-generating process does not match the model,

4. a function $h$ *is* structurally identifiable (i.e., $\epsilon$-nearly with $\epsilon = 0$) according to our Definition 2, where $h$ is in some sense the "nearest" linear function to $f$.

We emphasize that in this first example, we do not exploit the $\epsilon$-nearness relaxation in our definitions, only the distinction between statistical and structural identifiability.

**Example 2: masked autoencoders under imperfect reconstruction**  Now, consider observations generated by some arbitrary non-linear mixing of latent factors $X = g(Z)$. For a masked autoencoder, the following facts *could* be true all at once, such as in the case of imperfect reconstruction:

1. the internal representations given by the encoder $f$ are statistically identifiable according to our Definition 1,

2. the true data-generating process $g$ is not structurally identifiable according to Definition 2 because of imperfect reconstruction, but

3. some approximation to the data-generating process $h$ *is* structurally identifiable, but $h \neq g$ because masked training prevents perfect reconstruction, so

4. the model is not necessarily absolutely identifiable according to Reizinger et al. (2025a), depending on the assumptions on $h$ and $g$ (i.e. they might not reside in the same set of possible models).

In this case, we might not have the simple analytical form for $h$ that we do in Example 1, but it's clear that *something* about the data-generating process is structurally identifiable. We emphasize that our theory does not cover this case, because it requires a notion of "closeness" between $h$ and $g$ which is not necessarily obvious (nor is it necessarily covered by our conception of $\epsilon$-nearness, because the similarity is in observation space, not representation space).

In particular, Theorem 1 does not require perfect reconstruction to yield statistical identifiability (and there perhaps is structural identifiability of some process $h$ which is the nearest near-isometric approximation to the manifold of masked inputs, although we don't make this argument explicit), but Theorem 3 does require perfect reconstruction to yield structural identifiability of $g$. A similar observation can be made about statistical versus structural identifiability in exponential family models like GPTs, as discussed in Appendix A.3.1.

## A.2 THEORY ROADMAP

We provide a brief table of contents to this set of appendices, which cover our theoretical contributions.

- In Appendix A.3.1, we show the utility of our Definition 1 by showing that the statistical identifiability result from Roeder et al. (2021) meets our definition of statistical identiifaiblity for the penultimate layer of exponential family models such as GPTs.

- In Appendix A.3.2, we prove Theorem 1, our key statistical near-identifiability result up to $\mathcal{H}_{\text{rigid}}$ for the internal representations of general self-supervised models.

- In Appendices A.3.3 and A.3.4 we provide model-specific treatments for masked autoencoders and GPTs, showing that Theorem 1 holds for these models.

- In Appendix A.3.5, we prove Theorem 2, showing that linear ICA can resolve the rigid indeterminacy left by Theorem 1 (or indeed any other linear identifiability result), yielding statistical near-identifiability up to $\mathcal{H}_\sigma$ (the space of signed permutations).

- In Appendix A.3.6, we prove Theorem 3, showing that for bi-Lipschitz data-generating processes, statistical near-identifiability extends to structural near-identifiability.

- In Appendix A.4, we show that the dynamical isometry condition used for characterizing practical neural network training regimes implies the bi-Lipschitz assumption necessary for Theorem 1.

- In Appendix A.5, we prove that structural identifiability (Definition 2) is strictly stronger than statistical identifiability (Definition 1), including in the $\epsilon$-near case provided the transformation class $\mathcal{H}$ is bounded.

A.3    PROOFS

A.3.1    STATISTICAL NEAR-IDENTIFIABILITY OF EXPONENTIAL FAMILY MODELS

The goal of this section is to contextualize two important statistical identifiability results (Roeder et al., 2021) which apply to the penultimate layer of exponential family models, discussed in Section 3.1 ("Connection with prior results"). Specifically, we show that the result of Roeder et al. (2021) meets the requirements of our Definition 1, and explain how the result of Nielsen et al. (2025) relates to our definitions. In particular, we are interested in models which have losses taking the form in Equation 1 from Section 3.1:

$$\mathcal{L}_\theta(x, y) = -\boldsymbol{\eta}_\theta(x)^\mathsf{T} \mathbf{t}_\theta(y) + A_\theta(x) = -\log q_\theta(y \mid x)$$

where $\theta$ are the parameters of the model and $\boldsymbol{\eta}_\theta$ give the representations of interest. They key result of Roeder et al. (2021) hinges on on the assumption of sufficient diversity, which in the case of next-token predictors is the condition that the final linear classification head is in some sense "full rank".

First, we prove a simple lemma showing that sufficient diversity can be recast as a mild property of the data distribution, combined with an assumption on the model.

**Definition A.3.** *The mapping* $\mathbf{t} : \mathcal{Y} \to \mathbb{R}^D$ *satisfies sufficient diversity with respect to* $\mathbf{P}(x, y)$ *if repeated sampling from the marginal distribution* $y_d \sim \mathbf{P}(y)$ *yields* $D$ *linearly independent vectors* $\{\mathbf{t}(y_d) - \mathbf{t}(y_0)\}_{d=1}^D$ *for some* $y_0 \in \mathcal{Y}$.

**Lemma A.1.** $\mathbf{t} : \mathcal{Y} \to \mathbb{R}^D$ *satisfies the sufficient diversity assumption with respect to* $\mathbf{P}(y)$ *if and only if* $Cov[t(y)] \succeq \eta$ *for some* $\eta > 0$.

*Proof.* Assume $\mathbf{t}$ satisfies the sufficient diversity assumption. Let $v \in \mathbb{R}^D$ be any nonzero vector. By linear independence, we have $v^T (t(y_d) - t(y_0)) \neq 0$ almost surely for any fixed $y_0$, and therefore $v^T t(y) \neq c$ almost surely for any constant $c \in \mathbb{R}$. Thus, $v^T \text{Cov}[t(y)] v = \text{Var}[v^T t(y)] > 0$ as required.

To see the other direction, note that any $D$ draws from the distribution $\mathbf{P}(t(y))$ are linearly independent almost surely by the positive definiteness of the covariance matrix. $\square$

To see why this condition on the data distribution is extremely mild, note that even if $\mathbf{P}(y)$ is a finite categorical distribution over labels, so long as there are at least $D$ of them, it is possible that $\mathbf{t}$ satisfies sufficient diversity. Therefore, it is best regarded as a condition on the model. Now, we can state the key result of Roeder et al. (2021) as a lemma.

**Lemma A.2.** *(Theorem 1 of Roeder et al. (2021)) Let* $\mathbf{P}(x, y)$ *be a data distribution, and let* $\mathcal{M} = \{\mathcal{L}_\theta(x, y) = -\boldsymbol{\eta}_\theta(x)^\mathsf{T} \mathbf{t}_\theta(y) + A(x) : \theta \in \Theta\}$ *be a model with an exponential family loss. Then, if* $\mathcal{L}_\theta = \mathcal{L}_{\theta'}$ *almost everywhere with respect to* $\mathbf{P}$, *we have that* $\boldsymbol{\eta}_\theta(x) = L\boldsymbol{\eta}_{\theta'}(x)$ *almost everywhere with respect to* $\mathbf{P}$ *for some* $L \in \mathcal{H}_{linear}$ *whenever* $\mathbf{t}_\theta$ *and* $\mathbf{t}_{\theta'}$ *satisfy Definition A.3.*

The only additional requirement to meet our definition of $\epsilon$-near-identifiability (Definition 1) is that $\Theta$ is sufficiently rich to approximate a unique minimum of the loss. We make this concrete in the following definition. Importantly, this assumption is *not* equivalent to requiring that the true data distribution is in the model. Instead, it amounts to an assumption that the pointwise projection of the true conditional $\mathbf{P}(y \mid x)$ onto the space of exponential family distributions is unique and in the model. It turns out that this projection exists and is unique under the sufficient diversity assumption, together with very mild assumptions on the data distribution. Therefore, this parallels the situation of statistical identifiability of masked autoencoders (see Appendix A.3.3 and Appendix A.1, Example 2), where perfect reconstruction is not required for statistical identifiability (but might be required for a notion of structural identifiability).

**Definition A.4.** *For an exponential family loss* $\mathcal{L}_\theta(x, y) = -\boldsymbol{\eta}_\theta(x)^\mathsf{T} \mathbf{t}_\theta(y) + A_\theta(x)$, *a data distribution* $\mathbf{P}(x, y)$, *and a sufficient statistics function* $\mathbf{t}$ *satisfying sufficient diversity, consider the following minimization problems:*

$$\kappa^*(x) = \arg \min_{\kappa \in \mathcal{K}} KL(\mathbf{P}(y \mid x) \mid\mid \mathbf{Q}_\kappa)$$

*where* $\mathbf{Q}_\kappa(y) = \kappa^\mathsf{T} \mathbf{t}(y) + A(\kappa)$ *is the approximating distribution. Then, a parameter space* $\Theta$ *of an exponential family model of the form in Equation 1 is* sufficiently rich to exactly model the minimizer

*if this minimizer exists for every $x$, is unique for almost every $x$ with respect to $\mathbf{P}(x)$, and there exists $\theta \in \Theta$ such that $\boldsymbol{\eta}_\theta(x) = \kappa^*(x)$.*

**Remark.** For categorical likelihoods, the only requirement on the true data distribution $\mathbf{P}(y \mid x)$ is that all categories must have positive probability for the above to hold (for almost every $x$). This means that the assumption is trivially satisfied for e.g. GPT-class models assuming that any token might be next for any given state. This condition is called minimality (Wainwright & Jordan, 2008).

Now we give the full statement of the proposition showing that the Lemma A.2 combines with the above assumption to yield $\epsilon$-near-identifiability in the limit according to our definition (with $\epsilon = 0$).

**Proposition A.1.** *Let $\mathbf{P}(x, y)$ be a data distribution, and let $\mathcal{M} = \{\mathcal{L}_\theta(x, y) = -\boldsymbol{\eta}_\theta(x)^\intercal \mathbf{t}_\theta(y) + A_\theta(x) : \theta \in \Theta\}$ be a model with an exponential family loss. Then, for $F : \theta \mapsto \boldsymbol{\eta}_\theta$, $(\mathbf{P}, \Theta, \mathcal{L}_\theta, F)$ is indentifiable in the limit up to $\mathcal{H}_{linear}$ when the label distribution $\mathbf{P}(y)$ is sufficiently diverse and the approximating class $\Theta$ is sufficiently rich according to Definition A.4.*

*Proof.* Minimizing the expected loss of the model via empirical risk minimization is equivalent to minimizing the following cross-entropy:

$$\arg \min_{\theta \in \Theta} \mathbb{E}_{x,y \sim \mathbf{P}}[\mathcal{L}_\theta(x, y)] = \arg \min_{\theta \in \Theta} \mathbb{E}_{x,y \sim \mathbf{P}}[-\log q_\theta(y \mid x)]$$

$$= \arg \min_{\theta \in \Theta} \mathbb{E}_{x \sim \mathbf{P}}[\mathrm{CE}(q_\theta(y \mid x) \parallel \mathbf{P}(y \mid x))]$$

where $-\log q_\theta(y \mid x) \propto -\boldsymbol{\eta}_\theta(x)^\intercal \mathbf{t}_\theta(y) + A_\theta(x)$ parameterizes an exponential family approximating class. By sufficient richness, call $q_{\theta^*}$ the unique distribution induced by any minimizer $\theta^*$. So, for any two minimizers $\theta, \theta'$ we have $\mathcal{L}_\theta = \mathcal{L}_{\theta'}$ almost everywhere and Lemma A.2 yields identifiability in the limit (Definition 1, see Remark) as required. $\square$

Nielsen et al. (2025) extends this result to the case where it is not guaranteed that $q_\theta = q_{\theta'}$ for any two optimizers $\theta, \theta'$. For example, this might arise due to the sufficient richness condition not being met, or due to imperfect optimization. Instead, a closeness condition can be placed on $q_\theta$ and $q_{\theta'}$, yielding a notion that is similar to our notion of statistical $\epsilon$-near-identifiability. However, we note that Nielsen et al. (2025) does not aim to characterize the optima of a particular loss, instead relaxing this requirement to cover any procedure yielding likelihoods. This makes it an interesting extension of Roeder et al. (2021) in a different direction than ours.

### A.3.2 RIGID NEAR-IDENTIFIABILITY OF MODELS WITH BI-LIPSCHITZ MAPPINGS

We begin by briefly outlining the approach for proving Theorem 1. The key idea is to leverage modern results in isometric approximation (Vaisala, 2002; Alestalo et al., 2001). In particular, consider the latent spaces of two models which produce the same outputs. Call the mappings from latents to outputs in these two models $g$ and $g'$. If the mapping from latent to output is invertible, we can "stitch together" the latent spaces of the two models by a single function $g^{-1} \circ g'$. So, $g^{-1} \circ g'$ **maps the representations of an input under one model to its representations in the other model**. This proof technique was used in e.g. Zimmermann et al. (2021), where one of the latent spaces was the "true" latent factor space.

We assume that both $g$ and $g'$ are locally bi-Lipschitz, meaning that distances between nearby points are not too badly deformed, and that $g$ and $g'$ are smooth $C^1$ diffeomorphisms. When this is the case, $g^{-1} \circ g'$ is also locally bi-Lipschitz and smooth, and therefore nearly an isometry, with "nearly" determined by the bi-Lipschitz constants. The convexity of the latent spaces yields global bi-Lipschitzness with the same bounds.

The isometric approximation theory outlined in (Vaisala, 2002; Alestalo et al., 2001) then tells us how far $g' \circ g^{-1}$ is from an actual isometry, and allows us to construct bounds accordingly. Isometries on $\mathbb{R}^D$ are rigid transformations, and together, these facts yield near-identifiability up to $\mathcal{H}_{\mathrm{rigid}}$. Alternative isometric approximation theory based on the Friesecke-James-Müller theorem (Friesecke et al., 2002) yields a similar notion of near-rigidity in $L_2$ rather than $L_\infty$, which has been used to prove similar results in nonlinear ICA (Buchholz & Schölkopf, 2024). Readers interested in this

class of results should note that Buchholz (2023) shows that identifiability of exact isometries (and more generally, conformal maps and orthogonal coordinate transforms) can be cast as the uniqueness of a particular system of partial differential equations involving the Jacobian and Hessian of the mapping $g^{-1} \circ g'$. In light of impossibility results and counterexample constructions for nonlinear ICA (Locatello et al., 2019; Gresele et al., 2022; Buchholz, 2023), it is perhaps unsurprising that the uniqueness of solutions to such PDEs are non-trivial. Indeed, parallels may also be found in differential geometry, where e.g. the constructions used to prove the Nash and related embedding theorems are highly non-unique. In particular, these theorems state that there always exists a smooth isometric embedding of a smooth manifold (say, a data manifold) in a vector space of sufficiently large dimension (say, a latent space), but the solutions are highly non-unique as the dimension of the embedding grows, and isometric identifiability is only guaranteed in general for manifolds of equal dimension.

Now, we introduce the language and machinery required to prove the results presented in the main text. We rely on the following definition of locally bi-Lipschitz, which allows us to treat latent-to-observable mappings which might induce manifold structure in the ambient space, allowing the taking of a tighter constant $L$. In what follows, $\|\cdot\|$ denotes the usual $\ell_2$ norm unless otherwise stated.

**Definition A.5.** *A function $f : \mathcal{Z} \to \mathbb{R}^N$ is locally $(1+L)$-bi-Lipschitz if for every $\mathbf{z} \in \mathcal{Z} \subset \mathbb{R}^D$, there exists an open neighbourhood $U_\mathbf{z} \ni \mathbf{z}$ such that for all $\mathbf{z}' \in U_\mathbf{z}$, we have*

$$\frac{1}{1+L}\|\mathbf{z} - \mathbf{z}'\| \le \|f(\mathbf{z}) - f(\mathbf{z}')\| \le (1+L)\|\mathbf{z} - \mathbf{z}'\|$$

When $L = 0$, distances are preserved exactly, a notion referred to as *isometry*. When the bi-Lipschitz constraint is global (i.e., holds for any $U \subset \mathcal{Z}$) and $\mathcal{Z}$ is bounded, there is a nice relationship between the bi-Lipschitz property and the notion of a *near-isometry*, which allows for additive distortion of distances.

**Definition A.6.** *A function $f : \mathcal{Z} \to \mathbb{R}^N$ for $\mathcal{Z} \subset \mathbb{R}^D$ is an $\varepsilon$-near-isometry if for every $\mathbf{z}, \mathbf{z}' \in \mathcal{Z}$ we have $\|\mathbf{z} - \mathbf{z}'\| - \varepsilon \le \|f(\mathbf{z}) - f(\mathbf{z}')\| \le \|\mathbf{z} - \mathbf{z}'\| + \varepsilon$.*

In particular, if $f$ is globally $(1+L)$-bi-Lipschitz, we have that $f$ is also an $\varepsilon$-near-isometry where $\varepsilon = \Delta L$ where $\Delta = \sup_{\mathbf{z}, \mathbf{z}' \in \mathcal{Z}}\|\mathbf{z} - \mathbf{z}'\|$ is the diameter of $\mathcal{Z}$.

We require two lemmas to prove our main statistical identifiability result Theorem 1. The first is a fundamental result which shows that distance-preserving transformations on e.g. Euclidean spaces (or convex subsets thereof) are always rigid motions.

**Lemma A.3.** *(Mazur-Ulam Theorem, Russo (2017)) Let $\mathcal{Z}$ and $\mathcal{Z}'$ be closed, convex subsets of $\mathbb{R}^D$ with non-empty interior. Then, if $f : \mathcal{Z} \to \mathcal{Z}'$ is a bijective isometry, then $f$ is affine.*

We also leverage the following more recent result (see Vaisala (2002) for a history of isometric approximation) which shows that near-isometric mappings (such as globally $(1 + L)$-bi-Lipschitz functions on bounded domains) have bounded deviation from a truly isometric mapping.

**Lemma A.4.** *(Near-Isometries are Near Isometries, Theorem 2.2 of Alestalo et al. (2001)) Suppose $\mathcal{Z}, \mathcal{Z}' \subset \mathbb{R}^D$ with $\mathcal{Z}$ compact and $f : \mathcal{Z} \to \mathcal{Z}'$ is a $\Delta L$-near-isometry, where $\Delta = \sup_{\mathbf{z}, \mathbf{z}' \in \mathcal{Z}}\|\mathbf{z} - \mathbf{z}'\|$. Then, there exists an isometry $U : \mathbb{R}^D \to \mathbb{R}^D$ such that $\sup_{\mathbf{z} \in \mathcal{Z}}\|U(\mathbf{z}) - f(\mathbf{z})\| \le c_D\sqrt{L}\Delta$.*

With these two tools in hand, we can provide an intuitive overview of the assumptions required and a concrete statement of Theorem 1, proving statistical near-identifiability of the representations.

**Assumption Summary.** We assume that the representations have convex support (i.e., the pushforward of $\mathbf{P}(x)$ by any encoder $f_\theta$ has convex support). We assume that any "decoder" $g_\theta$ is injective, smooth (in particular, at least $C^1$), and is locally bi-Lipschitz with constant $1+L$. Note that injectivity, smoothness, and local bi-Lipschitzness of $g_\theta$ (with any constant) implies that $g_\theta$ is diffeomorphic onto its image.

**Theorem.** *Let $\mathbf{P}(x)$ be a data distribution, and let $\mathcal{M}$ be a model with a parameter space $\Theta$. Let $F : \theta \mapsto f_\theta$, $G : \theta \mapsto g_\theta$ and $H : \theta \mapsto g_\theta \circ f_\theta$, where $g_\theta$ is a smooth diffeomorphism and the pushforward of $\mathbf{P}$ through $f_\theta$ has convex support. Then, if $(\mathbf{P}, \Theta, \mathcal{L}_\theta, H)$ is statistically identifiable in the limit, then $(\mathbf{P}, \Theta, \mathcal{L}_\theta, F)$ is statistically $\epsilon$-near identifiable in the limit up to $\mathcal{H}_{rigid}$*

*for $\epsilon = c_D \sqrt{2L + L^2} \Delta$ where $1 + L$ is a local bi-Lipschitz constant bound for $g_\theta$, and $c_D$ and $\Delta$ are constants independent of the model (and L).*

*Proof.* Let $\theta$ and $\theta'$ be optima of the infinite-data limit of the empirical risk minimization problem for $\mathcal{M}$ with respect to $\mathbf{P}$.

Without loss of generality, assume that $f_\theta(\mathbf{x}) = 0 = f_{\theta'}(\mathbf{x})$ for some $\mathbf{x} \in \text{supp } \mathbf{P}$, noting that translating the latent spaces of both models do not alter any of our hypotheses. Then, by the identifiability of $(\mathbf{P}, \Theta, \mathcal{L}_\theta, H)$, we have that $T = g_{\theta'}^{-1} \circ g_\theta$ satisfies $T(0) = 0$, is clearly smooth and diffeomorphic onto its image, and is therefore globally $(1 + L)^2$-bi-Lipschitz by the convexity and compactness of the latent spaces and is a $\Delta(L^2 + 2L)$-near-isometry where $\Delta = \sup_{\mathbf{x},\mathbf{x}' \in \mathcal{X}} ||f_\theta(\mathbf{x}) - f_\theta(\mathbf{x}')||$ is the diameter of the latent manifold. Furthermore, we have that $T \circ f_\theta = f_{\theta'}$. By the above fact about bi-Lipschitz functions and Lemma A.4, we have that there exists an isometry $U$ such that

$$\underset{p(x)}{\text{ess sup }} ||f_{\theta'}(\mathbf{x}) - U(f_\theta(\mathbf{x}))||_2 = \underset{p(x)}{\text{ess sup }} ||T(f_\theta(\mathbf{x})) - U(f_\theta(\mathbf{x}))||_2$$

$$\leq c_D \sqrt{2L + L^2} \Delta$$

where $c_D$ is a constant depending on $D$. By convexity of the latent spaces and the Mazur-Ulam theorem for convex bodies (Lemma A.3), $U \in \mathcal{H}_{\text{rigid}}$ as required. $\square$

**Remark.** *The constant $c_D$ depends only on the latent dimension $D$ and can be computed by the following recursive formulae given in Alestalo et al. (2001):*

$$\varrho_1 = 3.3$$
$$\tau_1 = 6.2$$
$$\gamma_1(t)^2 = 0.1 + (t + \sqrt{t^2 + 6.2})^2$$
$$\varrho_{n+1}(\lambda) = 3.02 + \tau_n(\lambda)\sqrt{1 + \tau_n(\lambda)/\lambda^2} + \sum_{k=1}^{n} \varrho_k(\lambda)(2 + \varrho_k(\lambda)/\lambda^2)$$
$$\tau_{n+1}(\lambda) = \tau_n(\lambda) + \varrho_{n+1}(\lambda)(2 + \varrho_{n+1}(\lambda)/\lambda^2)$$
$$\gamma_{n+1}(t) = \min\{\max\{\gamma_n(\lambda), \beta_{n+1}(\lambda, t)\} : \lambda > 0\}$$
$$\beta_{n+1}(\lambda, t)^2 = 0.1 + (t + \sqrt{t^2 + \tau_{n+1}(\lambda)})^2 + \lambda^{-2} \sum_{k=2}^{n+1} \varrho_k(\lambda)^2$$

*with $c_D = \gamma_D(0)$ for any D. As an example, $c_3 \approx 18.8$.*

### A.3.3 MODEL-SPECIFIC RESULTS: MASKED AUTOENCODERS

The following assumptions are sufficient for a masked autoencoder to meet the criteria of Theorem 1.

**Model.** A **masked autoencoder** consists of a distribution $\mathbf{P}(m)$ defined over a space of masking functions $\mathcal{N} = \{m : \mathcal{X} \to \tilde{\mathcal{X}}\}$, an encoder $f_\theta : \mathcal{X} \cup \tilde{\mathcal{X}} \to \mathbb{R}^D$, and a decoder $g_\theta : \mathbb{R}^D \to \mathbb{R}^N$, where we assume $\mathcal{X} \subset \mathbb{R}^N$ is compact. For convenience, we write $h = g_\theta \circ f_\theta$. The loss function with respect to a data point $x$ and mask $m$ is $\mathcal{L}_{\text{MAE}}(h; m, x) = ||h(m(x)) - x||^2$. We assume that $\tilde{\mathcal{X}}$ is sequentially dense in $\mathcal{X} \cup \tilde{\mathcal{X}}$, and that marginalizing the joint distribution over the masks and data $\mathbf{P}(m, x, m(x))$ yields a distribution $\mathbf{P}(m(x))$ with full support on $\tilde{\mathcal{X}}$. Further, we assume that at the optimum, $g_\theta$ is diffeomorphic and locally $(1 + L)$-bi-Lipschitz for some constant $L \geq 0$. Finally, we assume that the image of $\mathcal{X}$ pushed forward through the encoder is convex, and that the parameterization of $h$ is continuous on $\mathcal{X} \cup \tilde{\mathcal{X}}$ and $\mathcal{M}$ is sufficiently rich to contain the function which attains the optimal value of the loss, as derived below.

*Proof.* We begin by showing the existence and uniqueness of the conditional expectation operator $\mathbb{E}[x \mid \tilde{x}]$. By the fact that the masking function distribution $\mathbf{P}(m)$ is independent of the data, we

have that there is a well-defined conditional density $\mathbf{P}(x \mid \tilde{x})$ arising from $\mathbf{P}(\tilde{x} \mid x) = \int \mathbf{P}(m)\delta(\tilde{x} - m(x))\, dm$, where $\delta$ is the Dirac delta. By compactness of $\mathcal{X}$, $x$ is integrable and therefore the conditional expectation exists and is unique almost everywhere in $\tilde{\mathcal{X}}$. In particular, we have the following expression:

$$\mathbb{E}[x \mid \tilde{x}] = \frac{\int_{\mathcal{X}} x\mathbf{P}(x)\mathbf{P}\left[m(x) = \tilde{x}\right]\, dx}{\int_{\mathcal{X}} \mathbf{P}(x)\mathbf{P}\left[m(x) = \tilde{x}\right]\, dx} \tag{2}$$

Then under the masked autoencoding model, $\hat{f}(\tilde{x}) = \mathbb{E}[x \mid \tilde{x}]$ is the unique minimizer of $\mathcal{L}_{\text{MAE}}$ by the standard fact that the conditional mean estimate minimizes the mean squared error loss. By the continuity of $\hat{f}$ on $X \cup \tilde{X}$ and sequential density of $\tilde{\mathcal{X}}$ in $\mathcal{X} \cup \tilde{\mathcal{X}}$, consider an arbitrary sequence in $\tilde{\mathcal{X}}$, $\tilde{x}_n \to x \in \mathcal{X}$, noting that $\hat{f}(x) = \lim_{\tilde{x}_n \to x} \hat{f}(\tilde{x}_n)$ is well-defined, unique, and agrees regardless of the choice of sequence. Note that this continuity is by hypothesis. We remark that it would be of interest to study properties of $\mathbf{P}(m)$ which lead to conditional expectation operators with certain properties including continuity. $\square$

**Remark.** *We emphasize that any such minimizer is also a minimizer when an arbitrary subsetting operation applied to both the prediction and the target, such as the usual subsetting to masked tokens in the loss for computational efficiency.*

### A.3.4 MODEL-SPECIFIC RESULTS: GPTs & SUPERVISED LEARNERS

The results for earlier-layer representations of GPTs and supervised learners take advantage of the penultimate-layer identifiability result in Proposition A.1. We formalize this with the assumptions below.

**Model.** Let $\mathbf{P}$ and $\mathcal{M}$ be a model satisfying Proposition A.1, with $F : \theta \mapsto f_\theta$ yielding the representation function of interest, $G : \theta \mapsto g_\theta$ yielding the map between the outputs of $f_\theta$ and the penultimate layer. Let $1 + L$ be a bound on the local bi-Lipschitz constant of the smooth diffeomorphism $g_\theta$, and suppose the pushforward of $\mathbf{P}$ through $f_\theta$ has convex support.

*Proof.* By Proposition A.1, we have statistical identifiability in the limit up to $\mathcal{H}_{\text{linear}}$ of $(\mathbf{P}, M, H)$ for $H : \theta \mapsto g_\theta \circ f_\theta$. Thus, Theorem 1 applies. $\square$

### A.3.5 INDEPENDENT COMPONENTS ANALYSIS IN LATENT SPACE

We consider the class of independent components analysis algorithms which optimize a contrast function, such as fastICA (Hyvärinen & Oja, 2000). In general, fastICA enjoys good convergence properties despite a lack of guarantees, and even in the face of mis-specification (Castella et al., 2013). Of particular concern in our setting is mis-specification of the kind such that there exists no orthogonal transformation in which the components are truly independent. This assumption is non-trivial to assess, and furthermore sufficient but not necessary for convergence (Castella et al., 2013), so we opt for something more relaxed, aimed at practical optimization. Specifically, we show that near-isometries are simple enough functions that they do not substantially alter the recovered components when ICA converges well, in the sense that they preserve well-differentiated optima.

**Model.** For an encoder $f$ with whitened outputs $\hat{f}(x)$, independent components analysis consists of maximizing the contrast $\mathcal{J} = \sum_{d=1}^{D} J(\mathbf{q}_d^T \hat{f}(x))$ where $J$ is a contrast function, over $(\mathbf{q}_1, \ldots, \mathbf{q}_D)^T = Q \in \mathbb{SO}(D)$, the space of special orthogonal matrices. As a technical condition, we require $\sup_f \|\text{Cov}[f(x)]\| \geq \lambda$ for some $\lambda > 0$, where the supremum is over any possible encoder (latents must have bounded correlation). The contrast function $J$ must be $C^2$ with $|J'(y)| \leq L_1$ and $|J''(y)| \leq L_2$ for rigid transformations of elements of a convex body $y = Uf(x)$. The optima of the ICA objective $\mathcal{J}$ must also be locally convex for $f(x)$, in the sense that for sufficiently large samples the Riemannian Hessian $\text{Hess}_Q \mathcal{J} \succcurlyeq \mu$ for some $\mu > 0$ at the optimum, under any perturbation of the data not larger than $\varepsilon$. Finally, the only indeterminacy to the optima of an individual ICA problem must be the usual invariances: signed permutations in $\mathcal{H}_\sigma$.

Under these conditions, we can show that $\varepsilon$-nearness does not introduce any further complications to the identifiability of the ICA model. We begin with a brief lemma about the stability of the whitening operation which typically precedes contrast-based ICA.

**Lemma A.5.** *Let $X$ and $X'$ be zero-mean random vectors in $\mathbb{R}^D$ such that $\|X - X'\| \leq \varepsilon$ almost surely and $\|X\|, \|X'\| \leq a$ almost surely. Furthermore, suppose the smallest eigenvalues of the covariance of $X$ and $X'$ are bounded below by $\lambda > 0$. Let $W = \Sigma^{-1/2}$ and $W' = \Sigma'^{-1/2}$ be the usual whitening matrices for $X$ and $X'$ respectively. Then $\|W'X' - WX\| \leq C\varepsilon$ almost surely where $C = \lambda^{-1/2}(1 + \lambda^{-1}a^2)$.*

*Proof.* First, note that we have the following bound almost surely:

$$\|XX^T - X'X'^T\| \leq \|X(X - X')^T + (X - X')X'^T\|$$
$$\leq \|X\|\varepsilon + \|X'\|\varepsilon \leq 2a\varepsilon$$

Taking expectations yields $\|\Sigma - \Sigma'\| \leq 2a\varepsilon$. Now, using the resolvent equation $B^{-1} - A^{-1} = B^{-1}(B - A)A^{-1}$, we have the following bound on the difference in operator norms of $W$ and $W'$:

$$\|W' - W\| = \|\Sigma^{-1/2} - \Sigma'^{-1/2}\|$$
$$\leq \|W'\|\|\Sigma^{1/2} - \Sigma'^{1/2}\|\|W\|$$
$$\leq \frac{\|\Sigma' - \Sigma\|}{2\lambda^{3/2}}$$
$$\leq \frac{a\varepsilon}{\lambda^{3/2}}$$

where the second-to-last inequality follows from the fact that the matrix square root is $(1/(2\sqrt{\lambda}))$-Lipschitz when $\lambda > 0$. Operating on the desired norm directly:

$$\|W'X' - WX\| \leq \|W'X' - W'X + W'X - WX\|$$
$$\leq \|W'(X' - X)\| + \|(W' - W)X\|$$
$$\leq \frac{\varepsilon}{\sqrt{\lambda}} + \frac{a^2\varepsilon}{\lambda^{3/2}}$$

almost surely so the proposition holds. $\square$

The next lemma simplifies the exposition of our main theorem. We show that PCA resolves the first-order dependence structure (i.e. the non-rigid component of linear transformations), leaving only a rigid transformation to be resolved by ICA.

**Lemma A.6.** *Suppose there exists functions $f, f' : \mathcal{X} \to \mathcal{Z}$ such that for $p(x)$ supported on $\mathcal{X}$, we have $\mathbb{E}_{\mathbf{P}(x)}[f(x)] = \mathbb{E}_{\mathbf{P}(x)}[f'(x)] = 0$, full-rank covariance of $f(x)$ and $f'(x)$, and furthermore $\sup_{x \in \mathcal{X}} \|f(x) - Af'(x)\| \leq \varepsilon$ for some invertible matrix $A$ with positive determinant and $\varepsilon \geq 0$. Then, there exists $U \in \mathcal{H}_{rigid}$ such that $\sup_{x \in \mathcal{X}} \|\hat{f}(x) - U\hat{f}'(x)\| \leq C\varepsilon$ where $\hat{f}, \hat{f}'$ are the whitened outputs of $f$ and $f'$ respectively, $U \in \mathcal{H}_{rigid}$, and $C = \lambda_A^{-1/2}\lambda^{-1/2}\left(1 + \frac{\Lambda_A^2}{\lambda_A\lambda}a^2\right)$ where $\Lambda_A$ and $\lambda_A$ are the largest and smallest singular values of $A$ respectively, $\lambda$ is a lower bound on the smallest eigenvalues of the covariance of $f(x)$ and $f'(x)$, and $\|f(x)\|, \|f'(x)\| \leq a$ almost surely with respect to $\mathbf{P}(x)$.*

*Proof.* Denote and $W = \Sigma^{-1/2}$ and $W' = \Sigma'^{-1/2}$ the usual whitening matrices for $f$ and $f'$ respectively. Let $W_A$ be any whitening matrix for $Af'$. Let $U = W_A A W'^{-1}$. Then $U$ is orthogonal (and can be made to have determinant 1 with a sign flip by the freedom to choose $W_A$) because $UU^T = W_A A W'^{-1} W'^{-T} A^T W_A = W_A \Sigma'_A W_A^T = I$, and we have:

$$\|\hat{f}(x) - U\hat{f}'(x)\| = \|Wf(x) - W_A A W'^{-1} \hat{f}'(x)\|$$
$$= \|Wf(x) - W_A A f'(x)\|$$
$$\leq \lambda_A^{-1/2} \lambda^{-1/2} \left(1 + \frac{\Lambda_A^2}{\lambda_A \lambda} a^2\right) \varepsilon$$

almost surely with respect to $\mathbf{P}(x)$ where the final line follows by application of the previous lemma to the usual whitening matrices. The determinant of $U$ is positive by the fact that the usual whitening matrices can be made unique by selecting their positive definite forms and $A$ has positive determinant, so $U \in \mathcal{H}_{\text{rigid}}$. $\square$

**Lemma A.7.** *(Implicit Function Theorem, de Oliveira (2014) Theorem 2) Let $F \in C^1(\Omega; \mathbb{R}^m)$ where $\Omega \subset \mathbb{R}^N \times \mathbb{R}^m$ is open. Suppose there exists a point $(a, b) \in \Omega$ such that $F(a, b) = 0$ and $\frac{\partial F}{\partial y}(a, b)$ is invertible, where $y$ represents the part of the argument $f$ in $\mathbb{R}^m$. Then, there exists an open set $X \subset \mathbb{R}^n$ such that $a \in X$ and an open set $Y \subset \mathbb{R}^m$ such that $b \in Y$ and:*

- *For each $x \in X$, there is a unique $y = f(x) \in Y$ such that $F(x, f(x)) = 0$*

- *$f(a) = b$ and $f$ is $C^1$ with $Df(x) = -\left(\frac{\partial F}{\partial y}(x, f(x))\right)^{-1} \left(\frac{\partial F}{\partial x}(x, f(x))\right)$ for all $x \in X$*

Finally, this lemma shows the crux of our argument: if ICA converges well in any pair of latent spaces, the solutions can't differ too much.

**Lemma A.8.** *(Finite-Sample ICA Under Perturbations) Consider whitened observations $\{\mathbf{x}_n\}_{n=1}^N$ and corruptions $\{\varepsilon_n\}_{n=1}^N$ (such that the corrupted observations $\mathbf{y}_n = \mathbf{x}_n + \varepsilon_n$ are also whitened) both in $\mathbb{R}^D$ such that $\|\mathbf{x}_n\| \leq a$ and $\|\varepsilon_n\| \leq b$ for all $n = 1, \ldots, N$. Let $Q_\star$ denote a stationary point of the optimization problem*

$$\max_{Q \in \mathbb{SO}(D)} \frac{1}{N} \sum_{n=1}^N \sum_{d=1}^D J(\mathbf{q}_d^T \mathbf{x}_n)$$

*Then, there exists a stationary point $Q_\star(\varepsilon_1, \ldots, \varepsilon_n)$ of the perturbed optimization problem*

$$\max_{Q \in \mathbb{SO}(D)} \frac{1}{N} \sum_{n=1}^N \sum_{d=1}^D J(\mathbf{q}_d^T(\mathbf{x}_n + \varepsilon_n))$$

*such that $\|Q_\star \mathbf{x}_n - Q_\star(\varepsilon_1, \ldots, \varepsilon_N)(\mathbf{x}_n + \varepsilon_n)\| \leq C + b$ for all $N$ where $C = \frac{L_2(a+b) + \sqrt{D}L_1}{\mu} ab$. Furthermore, $PQ_\star(\varepsilon_1, \ldots, \varepsilon_N)$ is a stationary point of the perturbed optimization problem attaining the same value for any signed permutation matrix $P \in \mathcal{H}_\sigma$ such that $\det P = 1$.*

*Proof.* We treat $\mathbb{SO}(D)$ as a Riemannian manifold and consider the properties of the perturbed optimization problem. For notational convenience, write $\varepsilon = (\varepsilon_1, \cdots, \varepsilon_N)^T$ noting that then $\|\varepsilon\| \leq \sqrt{N}b$.

We consider first the Euclidean gradient with respect to $Q$, where we write the perturbed sample as $\mathbf{y}_n = \mathbf{x}_n + \varepsilon_n$. Denote $\mathbf{g}_n(Q) = \left(J'(\mathbf{q}_1^T \mathbf{y}_n), \ldots, J'(\mathbf{q}_D^T \mathbf{y}_n)\right)^T$ for any $n$. Then:

$$\nabla_Q \mathcal{J}(Q, \varepsilon) = \frac{1}{N} \sum_{n=1}^N \mathbf{g}_n(Q)\mathbf{y}_n^T$$

The tangent space at a point $Q \in \mathbb{SO}(D)$ can be parameterized by the vector space of skew-symmetric matrices. Denoting $\text{skew}(A) = \frac{1}{2}(A - A^T)$, the Riemannian gradient is given by the projection of $\nabla_Q \mathcal{J}$ onto the tangent space:

$$\operatorname{grad}_Q \mathcal{J}(Q, \varepsilon) = Q \operatorname{skew}(Q^T \nabla_Q \mathcal{J}(Q, \varepsilon))$$

With this machinery, we can apply the Euclidean implicit function theorem (IFT). Let $Q_\star \in \mathbb{SO}(D)$ be an unperturbed optimum. Adopt the matrix exponential parameterization in a neighbourhood about $Q_\star$, i.e., let $\gamma : N \to \mathbb{SO}(D)$ be given by $\gamma(\Omega) = Q_\star \exp(\Omega)$ for some sufficiently small $N \subset \mathbb{R}^{D(D-1)/2}$ for the map to be injective. Let

$$F(\varepsilon, \Omega) = \operatorname{grad}_Q \mathcal{J}(\gamma(\Omega), \varepsilon) = \gamma(\Omega) \operatorname{skew}(\gamma(\Omega)^T \nabla_Q \mathcal{J}(\gamma(\Omega), \varepsilon))$$

and apply the IFT to $F$, noting that the hypotheses hold at $\Omega = 0$ and $\varepsilon = 0$. We thus have some neighbourhoods $E \subset \mathbb{R}^{N \times D}$ and $O \subset \mathbb{R}^{D(D-1)/2}$ such that for any $\varepsilon \in E$ there exists $\Omega_\star(\varepsilon) \in O$ such that

$$
\begin{aligned}
\|D_{\varepsilon_n} \Omega_\star(\varepsilon_n)\|_F &= \|\operatorname{Hess}_Q^{-1} \mathcal{J}(\gamma(\Omega_\star(\varepsilon)), \varepsilon)\| \left\| \frac{\partial F}{\partial \varepsilon}(\varepsilon, \Omega_\star(\varepsilon)) \right\| \\
&\leq \frac{1}{\mu} \left\| \operatorname{skew}\left( \gamma(\Omega)^T \frac{\partial \nabla_Q \mathcal{J}}{\partial \varepsilon_n} \right) \right\| \\
&\leq \frac{1}{\mu} \left\| \frac{\partial \nabla_Q \mathcal{J}}{\partial \varepsilon_n} \right\| \\
&\leq \frac{L_2(a+b) + L_1 \sqrt{D}}{\mu N}
\end{aligned}
$$

Here, the bound on the Hessian comes by hypothesis, while the second term follows from the fact that the skew operator and multiplication by an orthogonal matrix is norm-preserving, combined with the following expression for the Euclidean directional cross-derivative:

$$\partial_{\varepsilon_n} \nabla_Q \mathcal{J}[\mathbf{h}] = \frac{1}{N} \left( (R_n(Q) Q \mathbf{h}) \mathbf{y}_n^T + \mathbf{g}_n(Q) \mathbf{h}^T \right)$$

where $R_m(Q) = \operatorname{diag}\{J''(\mathbf{q}_1^T \mathbf{y}_m), \dots, J''(\mathbf{q}_D^T \mathbf{y}_m)\}$ and we have $\|\partial_{\varepsilon_n} \nabla_Q \mathcal{J}[\mathbf{h}]\| \leq \frac{L_2(a+b) + \sqrt{D} L_1}{N}$ as required. Aggregating across all $n$, we have $\|D_\varepsilon \Omega_\star(\varepsilon)\|_F \leq \frac{L_2(a+b) + L_1 \sqrt{D}}{\mu \sqrt{N}}$. By hypothesis, the additive symmetry between $\mathbf{x}_n$ and $\varepsilon_n$ is sufficient for this bound to hold for any sufficiently small perturbation.

As a result, we have

$$
\begin{aligned}
\|Q_\star(\varepsilon)(\mathbf{x}_n + \varepsilon_n) - Q_\star(0) \mathbf{x}_n\| &\leq \|Q_\star(\varepsilon) - Q_\star(0)\| \|\mathbf{x}_n\| + \|Q_\star(\epsilon) \varepsilon_n\| \\
&\leq \|\Omega_\star(\varepsilon) - \Omega_\star(0)\| \|\mathbf{x}_n\| + b \\
&\leq \frac{(L_2(a+b) + L_1 \sqrt{D}) ab}{\mu} + b
\end{aligned}
$$

as required. $\qquad\square$

Below, we provide a complete overview of the assumptions we make for Theorem 2.

**Assumption Summary.** We assume that the representation distribution (i.e. the pushforward of $\mathbf{P}(x)$ by any encoder $f_\theta$) has full-rank covariance, and that the support of the distribution has diameter bounded by $\Delta$. The eigenvalues of the covariance matrices are assumed to be bounded below by $\lambda$. Furthermore, we assume that the identifiability up to $\mathcal{H}_{\text{linear}}$ of $(\mathbf{P}, \Theta, \mathcal{L}_\theta, F)$ (where $F : \theta \mapsto f_\theta$) is satisfied for linear maps with singular values bounded between $\lambda_A$ and $\Lambda_A$. Finally, we assume that the contrast function used for ICA is Lipschitz (with constant $L_1$) and has Lipschitz derivative

(with constant $L_2$), and that ICA converges such that the Riemannian Hessian at the optimum has eigenvalues bounded below by $\mu > 0$.

With these lemmata, Theorem 2 in the main text follows easily.

**Theorem.** *Suppose $(\mathbf{P}, \Theta, \mathcal{L}_\theta, F)$ is $\epsilon$-nearly identifiable up to $\mathcal{H}_{linear}$ for $F : \theta \mapsto f_\theta$. Then, a new model $\mathcal{M}'$ which applies whitening and contrast function-based independent components analysis to the latent representations given by $f_\theta$ is $\epsilon'$-near-identifiable up to $\mathcal{H}_\sigma$ for $\epsilon' = K\epsilon + K'\epsilon^2$, where $K$ and $K'$ are constants free of $\epsilon$ that depends on the maximum diameter of the latent space $\Delta$, the spectra of the covariance matrix of the representations, and the properties of the ICA contrast function.*

*Proof.* Let $f_\theta$ and $f_{\theta'}$ be two optimal encoders. By $\epsilon$-near identfiability up to $\mathcal{H}_{\text{linear}}$, there exists $A$ such that $\text{ess sup}_{\mathbf{P}(x)}\|f_\theta(x) - Af_{\theta'}(x)\| \leq \epsilon$. If $A$ has positive determinant, Lemma A.6 with the usual whitening applied to both encoders yields $C\epsilon$-near-identifiability up to $\mathcal{H}_{\text{rigid}}$ for $C = \lambda_A^{-1/2}\lambda^{-1/2}\left(1 + \frac{\Lambda_A^2}{\lambda_A\lambda}a^2\right)$. If not, the sign of a single latent can be flipped, which flips the sign of a single column of $A$, allowing the application of Lemma A.6. Denote the whitened encoders by $\widehat{f_\theta}(x)$ and $\widehat{f_{\theta'}}(x)$. Lemma A.8 then applies with $a = \Delta$ and $b = C\epsilon$, where $\Delta$ is the maximum diameter of any latent space, yielding a bound $C' = \frac{L_2(\Delta+C\epsilon)+\sqrt{D}L_1}{\mu}\Delta C\epsilon$. Taking the limit as $N \to \infty$ and denoting $\widehat{f_\theta^{\text{ICA}}}(x)$ and $\widehat{f_{\theta'}^{\text{ICA}}}(x)$ the outputs of the encoders with whitening and ICA applied, we have $\text{ess sup}_{p(x)}\|\widehat{f_\theta^{\text{ICA}}}(x) - P\widehat{f_{\theta'}^{\text{ICA}}}(x)\| \leq K\epsilon + K'\epsilon^2$ for $K = \frac{L_2\Delta^2C+\sqrt{D}L_1\Delta C}{\mu}$ and $K' = \frac{L_2C^2\Delta}{\mu}$, and the indeterminacy $P \in \mathcal{H}_\sigma$ arises by the fact that any signed permutation of the latents is a maximum of the ICA objective, as required. $\qquad \square$

### A.3.6 Structural near-identifiability via ICA

Here, we are able to give a short proof of Theorem 3 by leveraging identical arguments to the proof of Theorem 2.

**Assumption Summary.** We assume that the representation distribution (i.e. the pushforward of $\mathbf{P}(x)$ by any encoder $f_\theta$) has full-rank covariance, and that the support of the distribution has diameter bounded by $\Delta$. The eigenvalues of the covariance matrices are assumed to be bounded below by $\lambda$. Furthermore, we assume that the identifiability up to $\mathcal{H}_{\text{linear}}$ of $(\mathbf{P}, \Theta, \mathcal{L}_\theta, F)$ (where $F : \theta \mapsto f_\theta$) is satisfied for linear maps with singular values bounded between $\lambda_A$ and $\Lambda_A$. Finally, we assume that the contrast function used for ICA is $C^2$ with Lipschitz first (with constant $L_1$) and second (with constant $L_2$) derivatives, and that ICA converges such that the Riemannian Hessian at the optimum has eigenvalues bounded below by $\mu > 0$. Finally, the end-to-end model $g_\theta \circ f_\theta$ must reconstruct its inputs perfectly at the optimum and the true data-generating process $g$ must be locally $(1 + \delta)$-bi-Lipschitz, smooth, and injective (and therefore diffeomorphic onto its image), with the data-generating factors being white (i.e. zero mean, unit variance), being non-Gaussian and having independent components. Finally, we assume that $\Theta$ is sufficiently rich so that $u \in \mathcal{M}$, i.e., the model can approximate the ground-truth data-generating structure.

**Theorem.** *Let $\mathbf{P}(u)$ be some multivariate distribution with independent non-Gaussian components with zero mean and unit variance, and consider data $\mathbf{P}(x)$ generated by pushforward through a smooth diffeomorphism $g$ such that $g$ is locally $(1 + \delta)$-bi-Lipschitz. Let $\mathcal{M}$ be a model with a sufficiently rich parameter space $\Theta$. Let $F : \theta \mapsto f_\theta$, $G : \theta \mapsto g_\theta$ and $H : \theta \mapsto g_\theta \circ f_\theta$. Then, if $(\mathbf{P}, \Theta, \mathcal{L}_\theta, H)$ structurally identifies the identity function in the limit (i.e. attains perfect reconstruction), we have that $(\mathbf{P}, \Theta, \mathcal{L}_\theta, F)$ $\epsilon$-nearly identifies the structure $g^{-1}$ up to $\mathcal{H}_{rigid}$, and furthermore that a new model $\mathcal{M}'$ which applies whitening and independent components analysis to the latent representations given by $f_\theta$ $\epsilon'$-nearly identifies the structure $g^{-1}$ up to $\mathcal{H}_\sigma$ where $\epsilon$ and $\epsilon'$ depend on $\delta$ and Lipschitz bounds on $g_\theta$, and $\epsilon'$ depends additionally on the spectrum of the covariance matrix of the representations and the properties of the ICA contrast function employed.*

*Proof.* Take $\delta = \max\{\delta, L\}$ as the maximum of the two local bi-Lipschitz constants of the data-generating process and the bound on the local bi-Lipschitz constant of the decoders in the model class. Theorem 1 yields $\delta'$-near-identifiability in the limit up to $\mathcal{H}_{\text{rigid}}$ for $\delta' = c_D\sqrt{2\delta + \delta^2}\Delta$. With the inverse of the true data-generating map taking the place of one of the encoders, the same argument

from the proof of Theorem 1 yields the first claim, namely $\delta'$-near structural identifiability of $g^{-1}$ up to $\mathcal{H}_{\text{rigid}}$.

The rest of the proof follows similarly to the proof of Theorem 2 in Appendix A.3.5, with the inverse of the true data-generating map taking the place of one of the encoders. In particular, let $\theta$ be an optimum and consider $f_\theta$ and $g_\theta$. Without loss of generality, assume that $f_\theta(\mathbf{x}) = 0 = g^{-1}(\mathbf{x})$, noting that the encoder $f_\theta$ and decoder $g_\theta$ can be translated arbitrarily without altering any of our hypotheses. The same argument then applies, yielding the result. The precise constant is the same, with $\epsilon = K\delta' + K'\delta'^2$ for $K$ and $K'$ as defined in the statement of Theorem 2. $\qquad\square$

## A.4 Dynamical isometry and bi-Lipschitzness

**Proposition.** *Suppose $f : \mathbb{R}^D \to \mathbb{R}^N$ is once differentiable and satisfies dynamical isometry, in the sense that the singular values $\lambda_i$ of the Jacobian $J$ satisfy $|\lambda_i - 1| \le \epsilon$ for $i = 1, \ldots, \min\{D, N\}$ for some $1 > \epsilon \ge 0$. Then, $\mathbf{f}$ is locally $L$-bi-Lipschitz for $L = \frac{1+\epsilon}{1-\epsilon}$.*

*Proof.* Let $x, y \in \mathbb{R}^D$. Then the mean value theorem yields the bound $\|f(x) - f(y)\| \le \|J_f\|\|x - y\| \le (1 + \epsilon)\|x - y\|$. By the same argument, $(1 - \epsilon)\|x - y\| \le \|f(x) - f(y)\|$, and taking $L = \frac{1+\epsilon}{1-\epsilon}$ (a bound on the condition number of $J$) yields a suitable bi-Lipschitz constant. $\qquad\square$

**Remark.** *Many popular regularization techniques spanning architectures and tasks optimize implicitly or explicitly for dynamical isometry, with the level of evidence ranging from theoretical to empirical. For example, weight decay has been shown theoretically and empirically to do so (Zhang et al., 2019), normalization techniques in generative adversarial networks optimize for it directly (Karras et al., 2020; Miyato et al., 2018), residual layers yield this property (Bachlechner et al., 2020), and specialized techniques have been developed to yield it at initialization (Xiao et al., 2018).*

## A.5 Structural identifiability implies statistical identifiability

Below, we prove that statistical (near-)identifiability is implied by structural (near-)identifiability.

**Theorem.** *Suppose $(\mathbf{P}, \Theta, \mathcal{L}_\theta, F)$ $\delta$-nearly identifies the structure $u$ up to $\mathcal{H}$ for $F : \theta \mapsto f_\theta$. Then, $(\mathbf{P}, \Theta, \mathcal{L}_\theta, F)$ is $\epsilon$-nearly identifiable up to $\mathcal{H}$ for some $\epsilon \in \mathbb{R}$ provided that $\mathcal{H}$ is bounded by $C \in \mathbb{R}^+$ as in the sense of an operator norm.*

*Proof.* Let $\theta, \theta' \in \mathcal{S}$ be solutions to the minimization problem $\min_{\theta \in \Theta} \mathbb{E}[\mathcal{L}_\theta]$ where $\mathcal{M} = \{\mathcal{L}_\theta : \theta \in \Theta\}$. By structural identifiability, we have that there exist $h$ and $h'$ relating $\theta$ and $\theta'$ respectively to $u$. More concretely, we have:

$$\|h \circ f_\theta - u\|_{L^p} \le \delta$$
$$\|h' \circ f_{\theta'} - u\|_{L^p} \le \delta$$

Take $h^* = h^{-1} \circ h'$, where the inverse and composition are well-defined because $\mathcal{H}$ is a group of functions mapping $\mathbb{R}^D$ to itself. $h^*$ can be understood as mapping $f_{\theta'}$ as close to $f_\theta$ as possible under the available assumptions. Then, we have

$$
\begin{aligned}
\|f_\theta - h^* \circ f_{\theta'}\|_{L^p} &= \|h^{-1} \circ h \circ f_\theta - h^{-1} \circ h' \circ f_{\theta'}\|_{L^p} \\
&= \|h^{-1} \circ h \circ f_\theta - h^{-1} \circ u + h^{-1} \circ u - h^{-1} \circ h' \circ f_{\theta'}\|_{L^p} \\
&\le \|h^{-1}\|_{\text{op}} \left(\|h \circ f_\theta - u\|_{L^p} + \|h' \circ f_{\theta'} - u\|_{L^p}\right) \\
&\le 2C\delta
\end{aligned}
$$

where the third line follows by the triangle inequality, and the final line by structural identifiability of $u$ and boundedness of $\mathcal{H}$, yielding the result.

The constant $C$ arises because solutions might exist on a different scale from the data-generating process $u$ (and therefore from each other). Partly as a result of this fact, this theorem likely is not useful for certain identifiability classes such as $\mathcal{H}_{\text{linear}}$, which even if bounded by assumption can likely yield more fruitful identifiability results via a direct approach. On the other hand, for identifiability classes like $\mathcal{H}_{\text{rigid}}$, the bound is in some sense tight.

Finally, we emphasize that the result is largely agnostic to the choice of reasonable norms, although we have in mind the usual operator norm and an $L^p$ norm for $1 \leq p \leq \infty$, taken with respect to the data distribution $\mathbf{P}(x)$. □

**Remark.** *Taking $\delta = 0$ shows that structural identifiability implies statistical identifiability in the limit.*

## A.6 CONFORMAL MAPS AS NEAR-ISOMETRIES

In this section we make rigorous the arguments outlined in Section 3.3.1. First, we outline the mollification argument which allows us to take derivatives in $L^2$ without hassle.

**Mollification** Let $\phi_{\sigma^2}$ be the isotropic zero-mean Gaussian density with variance $\sigma^2$. For any image represented by a function $F$, note that $F_\sigma = F * \phi_{\sigma^2}$ is a "smooth" version of that image without hard edges. Accordingly, mollifying all images by the same Gaussian means that taking the Gateaux derivative of image-valued functions becomes possible in $L^2$, and only introduces a multiplicative constant dependent on the variance $\sigma^2$ which we would like to ignore. To see that this is possible, consider the data-generating process for a square articulating along the $x$-axis according to a coordinate $p$ outlined in Section 3.3.1:

$$f(p) = \left[(x, y) \mapsto \mathbf{1}_{|x-p| \leq r, |y| \leq r}\right] \in L^2([-1, 1])$$

Denote $f_\sigma(p) = f(p) * \phi_{\sigma^2}$. Then, convolving with the distributional derivative gives

$$f'_\sigma(p) = \mathbf{1}_{|y| \leq |r|} \left(\phi(x - p - r) - \phi(x - p + r)\right)$$
$$\|f'_\sigma(p)\|^2_{L^2} = \frac{2r}{\sqrt{\pi}\sigma} + \mathcal{O}(\exp(-\sigma^2))$$

Pick any two distinct latents $p_0$ and $p_1$. For any smoothed manifold, we have that their geodesic distance is given by $\left(\frac{2r}{\sqrt{\pi}\sigma} + \mathcal{O}(\exp(-\sigma^2))\right) |p_1 - p_0|$. The distance between any two points can then be "renormalized" against this distance by dividing through it, ensuring that as $\sigma \to 0$ what is left is a constant. This then implies that what we are actually computing in the subsequent sections is not a metric inherited from the $L^2$ norm at all, but rather defines a whole new geodesic distance on the limiting manifold of unsmoothed images. For notational clarity, we ignore mollification in the rest of this section and return to a heuristic argument for the next sections. We direct the interested reader to Grimes (2003), Section 2.6 for a fully rigorous treatment.

**Two-dimensional manifold** Now, we fully characterize the 2-dimensional manifold described in the main text. Let $\mathcal{Z} = \{(p, r) \mid a \leq p \leq b, R_0 \leq r \leq R\}$ denote the manifold of latent variables of the position of the square and its half-side length. Each point $\mathbf{z} \in \mathcal{Z}$ can be identified with an image:

$$f(p, r) = \left[(x, y) \mapsto \mathbf{1}_{|x-p| \leq r, |y| \leq r}\right] \in L^2([-1, 1])$$

**Directional derivatives** The derivative with respect to $p$ remains the same as in the main text, and the derivative with respect to $r$ follows similarly:

$$\|\partial_p f(p, r)\|^2 = 2r$$
$$\|\partial_r f(p, r)\|^2 = 8r$$

However, checking that $f$ is a conformal map also demands that $\partial_p f$ and $\partial_r f$ are orthogonal in $L^2$. To see this, note that finite difference approximation to $\partial_p f$ are the (negative) left and (positive) right edges of the square, while the finite difference approximation to $\partial r f$ are all (positive) edges of the square. Therefore, the top edges contribute nothing to the inner product $\langle \partial_p f, \partial_r f \rangle$ while the contributions of the left and right edges cancel. Thus, the Riemannian metric can be written

$$G(p, r) = 2r \begin{pmatrix} 1 & 0 \\ 0 & 4 \end{pmatrix}$$
$$G(p', r') = r' I$$

where the reparameterization $p' = p/4$ and $r' = 2r$ allows us to recover the isotropy.

**Near-isometry**   It remains to show how $f$ can be viewed as locally bi-Lipschitz. To see this, note that the we have the following global bound on the differential:

$$2R_0 \|\mathbf{u}\| \leq \|Df(\mathbf{u})\| \leq 2R\|\mathbf{u}\|$$

for $\mathbf{u}$ in the tangent space at any point along $\mathcal{Z}$. Accordingly, we have the following bound on the geodesic distance:

$$\sqrt{2R_0}\|\mathbf{z}_1 - \mathbf{z}_0\| \leq d_{\text{geo}}\left(f(\mathbf{z}_1), f(\mathbf{z}_0)\right) \leq \sqrt{2R}\|\mathbf{z}_1 - \mathbf{z}_0\|$$

where $d_{\text{geo}}$ is the geodesic distance along the image manifold. As a result (assuming w.l.o.g. that $R_0 \leq 1$), $f$ is locally $\sqrt{R/R_0}$-bi-Lipschitz. Furthermore, for any $f_\star^{-1}$ from the image manifold to a convex subset of $\mathbb{R}^2$ which is also $\sqrt{R/R_0}$-bi-Lipschitz, $f \circ f_\star^{-1}$ is globally $R/R_0$-bi-Lipschitz (with respect to the $\ell^2$ norm on both spaces, where the constant follows by convexity) and is therefore a near-isometry with constant $(R/R_0 - 1)\Delta$ where $\Delta = \sqrt{4(R - R_0)^2 + (a - b)^2}$ is simply the diameter of $\mathcal{Z}$.

## A.7   EXPERIMENTAL DETAILS

### A.7.1   WARMUP EXPERIMENT: MNIST

We conducted experiments on the MNIST dataset (LeCun et al., 2010) to validate our theoretical predictions regarding the identifiability of latent representations as a function of the bi-Lipschitz constant of the decoder.

**Training**   We trained pairs of orthogonal LeakyReLU autoencoders with the following architecture:

- **Encoder:** $\mathbb{R}^{784} \to \mathbb{R}^{784} \to \mathbb{R}^{784} \to \mathbb{R}^{784} \to \mathbb{R}^2$
- **Decoder:** $\mathbb{R}^2 \to \mathbb{R}^{784} \to \mathbb{R}^{784} \to \mathbb{R}^{784} \to \mathbb{R}^{784}$

All linear layers used orthogonal weight parametrization (no bias terms). LeakyReLU activations with leak constant $\alpha \in \{0.0, 0.25, 0.5, 0.75, 0.9, 1.0\}$ were applied at all intermediate layers. The latent dimension was $D = 2$. All models were fit using the Adam optimizer (Kingma & Ba, 2015) with learning rate $\eta = 5 \times 10^{-4}$ for up to 2000 epochs, minimizing the mean squared reconstruction error. Early stopping was applied with patience of 50 epochs and minimum improvement threshold of $10^{-6}$. Gradients were clipped to unit norm for stability. We repeated each configuration with 10 random seeds for robustness, yielding $6 \times 10 = 60$ experimental runs. We filtered experimental runs to exclude poorly converged autoencoders. Specifically, we removed runs where the reconstruction error

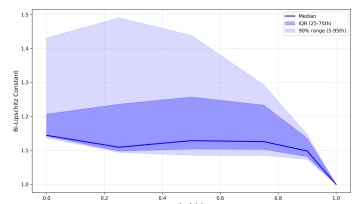

Figure A.3: The distribution of sample-level bi-Lipschitz constant estimates $B(z)$ tightens around 1 as $\alpha \to 0$.

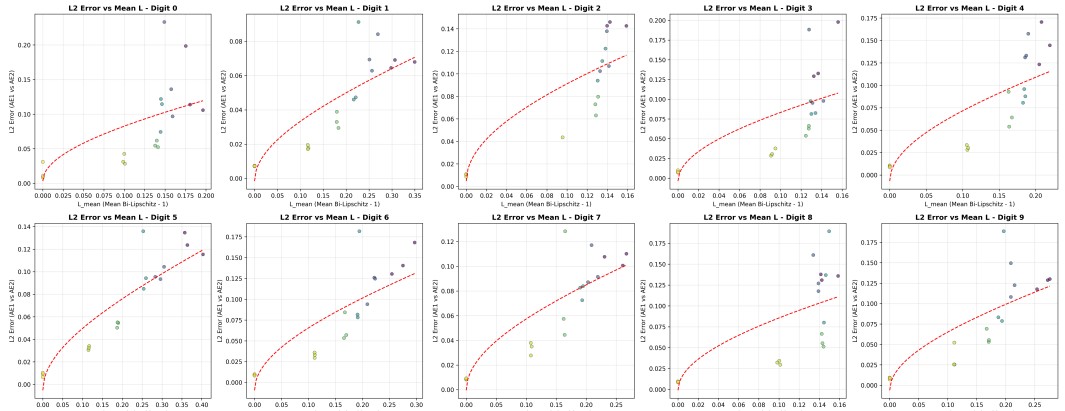

Figure A.6: In digit-specific models, controlling the bi-Lipschitz constant $L$ leads to improved identifiability (reduced $\ell_2$ error) with pattern similar to the full-dataset models.

of either autoencoder exceeded the 95th percentile observed at the reference leak value $\alpha = 0.9$. This removed 11/60 runs.

**Results** For a given autoencoder, we compute the representations $z$ of a random subset of 1000 samples. For the decoder $g$, the bi-Lipschitz constant at a given latent point $\mathbf{z}$ is given as

$$B(z) = \max_v \{\|J_g(z)v\|_2, 1/\|J_g(z)\|_2\}$$

where the maximum is taken over 10 randomly sampled unit vectors. We plot the distribution of $B(z)$ as a function of the leak constant $\alpha$ in Figure A.3. As $\alpha \to 1$, the distribution is typically well-concentrated around a mean not much larger than one, with a relatively small number of outliers. Figure 2 (in the main text) plots the samplewise estimate of $L = \mathbb{E}_{\mathbf{P}(z)}[B(z) - 1]$ versus identifiability. Although more robust to outliers, this is not a formal bound because Theorem 1 relies on a global $L$, i.e. $L = \max_{\mathbf{P}(z)}[B(z) - 1]$. For completeness, we plot using the maximum in Figure A.4, with the maximum taken over all 1000 samples. Both plots include fitted curves of the form $\ell_2$ error $= a\sqrt{L + L^2} + b$ to the identifiability measurements, consistent with the theorem.

As $\alpha \to 1$, the estimated bi-Lipschitz constants $L$ do indeed shrink toward 1 as expected. This suggests validity of the experimental testbed, but not Theorem 1 itself. However, as $L \to 1$, we see that identifiability does indeed improve significantly ($\ell_2$ error $\to 0$) as predicted by the Theorem. Notably, this is despite the fact that perfect reconstruction does not hold (Figure A.5).

We also completed the same experiment with a model per digit. This allows us to assess whether class-level indeterminacies play a role in identifiability in this problem setting like in (Nielsen et al., 2025). All hyperparameters and setup remains the same, except we fit three seeds per digit for a total of 10 digits $\times$ 6 leak values $\times$ 3 seeds $= 180$ runs. Filtering using the same rule for reconstruction removed 5/180 runs. Results are consistent for these per-digit autoencoders as well, suggesting that for this setting of hyperparameters, class-level indeterminacies do not play a substantial role in the level of empirical identifiability (Figure A.6).

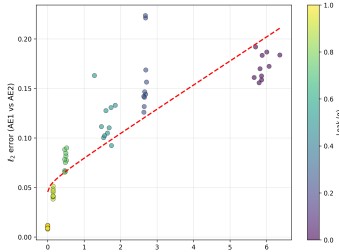

Figure A.4: Controlling the bi-Lipschitz constant $L$ leads to improved identifiability (reduced $\ell_2$ error). The proportionality does not appear to differ whether the max or mean bi-Lipschitz constant is estimated.

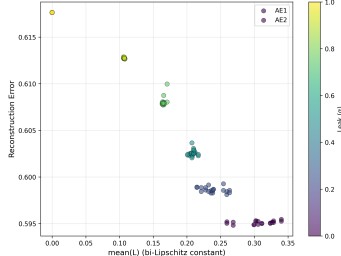

Figure A.5: Reconstruction error improves as the bi-Lipschitz constant grows (leak $\alpha \to 1$). Notably, poor reconstruction does not inhibit identifiability.

### A.7.2 MEASURING NEAR-IDENTIFIABILITY

We describe here the setup for the alignment experiments reported in Table 1. All reported statistics are *in-sample*, with representations taken from the same data on which the models were trained. For each model, we extract the following hidden states:

- **GPT-class models (Pythia-160M):** penultimate-layer hidden states.
- **MAE models:** latent representations obtained by averaging patch embeddings.
- **Supervised models (ResNet-18):** penultimate-layer representations.

We apply the following representation alignment techniques:

- **Permutation:** representation dimensions were matched using the Hungarian algorithm to resolve signed permutation indeterminacies.
- **Rigid:** estimated via the Procrustes algorithm with a global scaling constant (rigid similarity transform).
- **Linear:** estimated via least-squares regression with no additional regularization or constraints.
- **ICA:** estimated using FastICA (scikit-learn implementation), applied to the full representation dimension with all components retained. The Hungarian algorithm resolves the remaining signed permutation in latent space.

Alignment quality is reported as the mean per-example $\ell_2$ error, normalized by the *latent diameter*, defined as the maximum pairwise $\ell_2$ distance among representations. For ICA, efficiency is reported as the proportion of error reduction relative to the supervised rigid transform:

$$\text{ICA efficiency} = \frac{\text{Permutation} - \text{ICA}}{\text{Permutation} - \text{Rigid}}.$$

### A.7.3 DISENTANGLEMENT EXPERIMENTS

We reproduce only the autoencoder (AE) results, enough to validate that we obtain similar performance, and apply ICA to these models. Specifically, each hyperparameter setting is re-fit 3 times, and training step where the average performance is the best is selected (according to modularity, after filtering for reconstruction as in Hsu et al. (2023)).

As in the original experiments in Hsu et al. (2023), the latent spaces are overparameterized with the number of latents equal to twice the number of ground-truth sources $n_z = 2n_s$. As a result, the latent space is rank deficient. To avoid the introduction of additional hyperparameters for pruning inactive latents which could bias the performance of our approach (via e.g. whitening with dimensionality reduction), we fit the FastICA model without whitening. Inspection of the decoder Jacobian reveals that inactive latents are obvious (corresponding singular values are very near zero) and the remaining components of the Jacobian have similar scale (likely due to high weight decay), suggesting that full-rank whitening would have minimal impact here except to drive up the noise from inactive latents. As a result, we perform no whitening and preserve all latent dimensions.

Table 5 is an augmented version of Table 2, with standard errors computed across the 3 models with the best hyperparameter setting for each model. Results for all models other than AE (reproduction) and AE + ICA are quoted from Hsu et al. (2023), which averaged over 5 seeds instead of 3.

### A.7.4 OPENPHENOM EXPERIMENTS

The whitening and FastICA Hyvärinen & Oja (2000) algorithms are from the `scikit-learn` package Pedregosa et al. (2011). These models are trained at the patch level on the patches from the first channel of images in Rxrx3-core Kraus et al. (2025), where patches are specifically subsampled from the top left-hand corner. To ensure computational feasibility, a single patch is sampled from each image, yielding approximately 222K patches. Both models use the default hyperparameters (all principal components are kept and the contrast function is $\log \cosh$).

Gradient boosting models are trained using LightGBM Ke et al. (2017) (hyperparameters in Table 6). Models are trained to predict whether the image patch embedding is from an image which is perturbed or not. Models are evaluated using area under the receiver operator characteristic curve.

A separate model is trained on each plate, which is then evaluated on all remaining plates. Below, we report an augmented version of Table 3 with the standard error of the mean computed across folds. When more than one independent experiment (corresponding to a guide) is available for a given gene, the standard error is computed across all folds from all guides.

| Gene | Mean AUROC ± s.e. (↑) | | | | Sparsity ± s.e. (↑ more sparse) | | | |
|---|---|---|---|---|---|---|---|---|
| | Base | PCA | PCA + ICA | PCA + Rand | Base | PCA | PCA + ICA | PCA + Rand |
| CYP11B1 (1) | $0.663 \pm 0.008$ | $0.692 \pm 0.008$ | $\mathbf{0.709} \pm 0.005$ | $0.678 \pm 0.010$ | $0.184 \pm 0.006$ | $0.204 \pm 0.007$ | $\mathbf{0.237} \pm 0.006$ | $0.188 \pm 0.007$ |
| EIF3H (1) | $0.682 \pm 0.008$ | $0.724 \pm 0.004$ | $\mathbf{0.749} \pm 0.007$ | $0.725 \pm 0.003$ | $0.192 \pm 0.004$ | $0.224 \pm 0.010$ | $\mathbf{0.268} \pm 0.008$ | $0.214 \pm 0.006$ |
| HCK (1) | $0.670 \pm 0.012$ | $0.693 \pm 0.007$ | $\mathbf{0.711} \pm 0.005$ | $0.668 \pm 0.007$ | $0.156 \pm 0.004$ | $0.208 \pm 0.004$ | $\mathbf{0.241} \pm 0.006$ | $0.166 \pm 0.005$ |
| MTOR (6) | $0.663 \pm 0.003$ | $0.690 \pm 0.003$ | $\mathbf{0.705} \pm 0.003$ | $0.679 \pm 0.003$ | $0.166 \pm 0.002$ | $0.201 \pm 0.003$ | $\mathbf{0.233} \pm 0.003$ | $0.186 \pm 0.002$ |
| PLK1 (6) | $0.803 \pm 0.002$ | $0.811 \pm 0.002$ | $\mathbf{0.815} \pm 0.002$ | $0.792 \pm 0.002$ | $0.251 \pm 0.003$ | $\mathbf{0.307} \pm 0.004$ | $0.305 \pm 0.003$ | $0.262 \pm 0.003$ |
| SRC (1) | $0.660 \pm 0.004$ | $0.694 \pm 0.007$ | $\mathbf{0.706} \pm 0.004$ | $0.676 \pm 0.007$ | $0.170 \pm 0.006$ | $0.214 \pm 0.003$ | $\mathbf{0.240} \pm 0.008$ | $0.184 \pm 0.005$ |

In Table 7, we report an augmented version of Table 4.4 with the concentration scores, assessing sensitivity to the parameter $k$.

Table 5: Full results across datasets.

| **Shapes3D** | | | |
|---|---|---|---|
| **model** | **InfoM ↑** | **InfoE ↑** | **InfoC ↑** |
| AE | $0.41 \pm 0.03$ | $0.98 \pm 0.01$ | $0.28 \pm 0.01$ |
| $\beta$-VAE | $0.59 \pm 0.02$ | $0.99 \pm 0.02$ | $0.49 \pm 0.03$ |
| $\beta$-TCVAE | $0.61 \pm 0.03$ | $0.82 \pm 0.02$ | $0.62 \pm 0.02$ |
| BioAE | $0.56 \pm 0.02$ | $0.98 \pm 0.01$ | $0.44 \pm 0.02$ |
| AE (reproduction) | $0.36 \pm 0.05$ | $1.00 \pm 0.00$ | $0.18 \pm 0.06$ |
| AE + ICA | $0.78 \pm 0.02$ | $1.00 \pm 0.00$ | $0.42 \pm 0.09$ |
| *Discrete latent models* | | | |
| VQ-VAE | $0.72 \pm 0.03$ | $0.97 \pm 0.02$ | $0.47 \pm 0.03$ |
| VQ-VAE w/ weight decay | $0.80 \pm 0.01$ | $0.99 \pm 0.01$ | $0.46 \pm 0.02$ |
| QLAE | $0.95 \pm 0.02$ | $0.99 \pm 0.01$ | $0.55 \pm 0.02$ |

| **MPI3D** | | | |
|---|---|---|---|
| **model** | **InfoM ↑** | **InfoE ↑** | **InfoC ↑** |
| AE | $0.37 \pm 0.04$ | $0.72 \pm 0.03$ | $0.36 \pm 0.03$ |
| $\beta$-VAE | $0.45 \pm 0.03$ | $0.71 \pm 0.03$ | $0.51 \pm 0.03$ |
| $\beta$-TCVAE | $0.51 \pm 0.04$ | $0.60 \pm 0.04$ | $0.57 \pm 0.04$ |
| BioAE | $0.45 \pm 0.03$ | $0.66 \pm 0.04$ | $0.36 \pm 0.03$ |
| AE (reproduction) | $0.42 \pm 0.05$ | $0.66 \pm 0.28$ | $0.31 \pm 0.12$ |
| AE + ICA | $0.44 \pm 0.12$ | $0.66 \pm 0.28$ | $0.31 \pm 0.14$ |
| *Discrete latent models* | | | |
| VQ-VAE | $0.43 \pm 0.06$ | $0.57 \pm 0.04$ | $0.22 \pm 0.04$ |
| VQ-VAE w/ weight decay | $0.50 \pm 0.04$ | $0.81 \pm 0.04$ | $0.41 \pm 0.04$ |
| QLAE | $0.61 \pm 0.04$ | $0.63 \pm 0.05$ | $0.51 \pm 0.03$ |

| **Falcor3D** | | | |
|---|---|---|---|
| **model** | **InfoM ↑** | **InfoE ↑** | **InfoC ↑** |
| AE | $0.39 \pm 0.03$ | $0.74 \pm 0.03$ | $0.20 \pm 0.03$ |
| $\beta$-VAE | $0.71 \pm 0.05$ | $0.73 \pm 04$ | $0.70 \pm 0.03$ |
| $\beta$-TCVAE | $0.66 \pm 0.02$ | $0.74 \pm 0.04$ | $0.71 \pm 0.04$ |
| BioAE | $0.54 \pm 0.05$ | $0.73 \pm 0.04$ | $0.31 \pm 0.01$ |
| AE (reproduction) | $0.37 \pm 0.14$ | $0.75 \pm 0.01$ | $0.21 \pm 0.06$ |
| AE + ICA | $0.68 \pm 0.09$ | $0.75 \pm 0.01$ | $0.37 \pm 0.32$ |
| *Discrete latent models* | | | |
| VQ-VAE | $0.61 \pm 0.04$ | $0.83 \pm 0.05$ | $0.42 \pm 0.02$ |
| VQ-VAE w/ weight decay | $0.74 \pm 0.02$ | $0.86 \pm 0.04$ | $0.40 \pm 0.03$ |
| QLAE | $0.71 \pm 0.03$ | $0.77 \pm 0.02$ | $0.44 \pm 0.02$ |

| **Isaac3D** | | | |
|---|---|---|---|
| **model** | **InfoM ↑** | **InfoE ↑** | **InfoC ↑** |
| AE | $0.42 \pm 0.04$ | $0.80 \pm 0.02$ | $0.21 \pm 0.05$ |
| $\beta$-VAE | $0.60 \pm 0.03$ | $0.80 \pm 0.02$ | $0.51 \pm 0.03$ |
| $\beta$-TCVAE | $0.54 \pm 0.02$ | $0.70 \pm 0.02$ | $0.46 \pm 0.03$ |
| BioAE | $0.63 \pm 0.03$ | $0.65 \pm 0.03$ | $0.33 \pm 0.04$ |
| AE (reproduction) | $0.41 \pm 0.11$ | $0.80 \pm 0.05$ | $0.20 \pm 0.07$ |
| AE + ICA | $0.64 \pm 0.17$ | $0.80 \pm 0.05$ | $0.34 \pm 0.05$ |
| *Discrete latent models* | | | |
| VQ-VAE | $0.57 \pm 0.04$ | $0.87 \pm 0.05$ | $0.45 \pm 0.04$ |
| VQ-VAE w/ weight decay | $0.73 \pm 0.03$ | $0.81 \pm 0.03$ | $0.44 \pm 0.04$ |
| QLAE | $0.78 \pm 0.03$ | $0.97 \pm 0.03$ | $0.49 \pm 0.03$ |

| Hyperparameter | Value |
|---|---|
| is_unbalance | True |
| learning_rate | 0.05 |
| num_leaves | 31 |
| feature_fraction | 0.6 |
| reg_alpha | 5.0 |
| reg_lambda | 1.0 |
| min_gain_to_split | 0.8 |
| min_data_in_leaf | 30 |

Table 6: LightGBM hyperparameters used in all experiments.

Table 7: Sensitivity of concentration to $k$.

| Model | Concentration ($\uparrow$) | | |
|---|---|---|---|
| | $k = 25\%$ | $k = 33\%$ | $k = 50\%$ |
| Base (none) | 0.163 | 0.134 | 0.133 |
| PCA | 0.332 | 0.307 | 0.314 |
| PCA + ICA | 0.386 | 0.372 | 0.334 |
| PCA + RandRot | 0.287 | 0.280 | 0.235 |

