# OpenReview forum: "Statistical and structural identifiability in representation learning"
_ICLR.cc/2026/Conference — ICLR 2026 Poster_

### Official Review · Reviewer_p3of · 2025-10-27

**Soundness:** 3
**Presentation:** 1
**Contribution:** 3
**Rating:** 4
**Confidence:** 3

**Summary:**

This paper provides relaxed conditions for representational identifiability in self-supervised learning, extending previous results beyond the (pen)ultimate layer. For their weaker results, the authors proposed a further preprocessing step by linear ICA to remove the linear indeterminacy. The extension of identifiability to the approximate regime is an important contribution, especially for practial applicability of identifiability.

**The paper seems to be very interesting, and I believe it has great potential. However, I have concerns, especially about the imprecise use of he word "identifiability."** Indeed, what the authors call "recoverability" is what most of the literature calls identifiability.

Although I think this paper has the potential to be a spotlight/oral, given my current concerns, I chose a conservative score. **If the authors can address my concerns (or correct me if I have misunderstood any of their claims), then I will raise my score.**

**Strengths:**

- The paper aims to relax identifiability conditions, thus, increasing the practical relevance of the field of identifiability. Thus, it is very timely and useful.
- The extension to approximate identifiability is definitely underexplored (though not *entirely*, novel, see, e.g., [1])
- The theory is contextualized for practically relevant model families (MAEs/GPTs)
- The empirical evaluation is detailed (though a bit unclear, see weaknesses)
	- I especially liked the PCA+rand control in Table 3
- A.1.3-4 are nice discussions on architecture-specific cases of the theory
- A.1.5 is a nice discussion on ICA in latent space

- [ 1 ] Reizinger, Patrik, Szilvia Ujváry, Anna Mészáros, Anna Kerekes, Wieland Brendel, and Ferenc Huszár. "Position: Understanding LLMs Requires More Than Statistical Generalization." In Forty-first International Conference on Machine Learning.

**Weaknesses:**

My main concerns are about the clarity/phrasing, and NOT the technical contributions.

### Major points
- **My biggest concerns is the improper use of identifiability/recoverability**: when the authors refer to identifiability, they do not refer to the commonly used term in the ICA literature (which is formulated in terms of the ground-truth data generating process). The authors introduce (superfluously) recoverability for the (as far as I can tell) same concept. When they talk about identifiability, they refer to the definition used by Roeder et al., 2020. I strongly suggest to correct the use of the term to avoid confusion with the majority of the identifiability literature. Note that this does not devalue any contributions, only makes their presentation presumably confusing. For a comparison between the two identifiability notions, see, e.g., [ 1 ] (where the authors call it relative - Roeder - vs absolute - Hyvarinen - identifiability)
- Thm. 1.:
	- the theorem would benefit from a more-detailed explanation. In my opinion, the current intuition section is more like a sketch. At least I couldn't figure out the intuition.
	- E.g., why are all $F, G, H$ needed?
	- What is the role of bi-Lipschitzness?
	- Small remark: there is a identifiability result for next-token prediction, see, http://arxiv.org/abs/2503.08980
- Def. 2: rephrase the intuition with the mainstream identifiability definition (the connection between the two concepts in A.3 is nice)
- Table 1: it is missing context, it is unclear from the caption what the metrics mean, what was measured, etc. Please make the caption self-contained
- The proof of the Theorem in A.3 would benefit from more steps and inline explanations of what is happening between the steps

### Minor points
- L045: "Second, model-specific identifiability results fail to predict the empirical alignment seen across different architectures and modalities (Huh et al., 2024):"
	- This is unclear how it would hold for the most widely-used identifiability definition, as assumptions generally require assumptions that do not constrain the modality or the architecture. Please elaborate
- L053: In this paper, we argue that many current identifiability results in representation learning are overloaded with recoverability guarantees of some latent component of the data-generating process,
	- I don't understand this sentence
- L143: here, the "All or None" paper by Marconate et al. seems to be very relevant, please discuss it. Also, [ 1 ] proposes a simialr notion, termed $\varepsilon-$non-identifiability
- L277 - Example: I like the example, though it's a bit too involved and I am missing the intuition for it. If you could add such an explanation (preferably with a figure), that'd help the reader
- L363: the citation showing Anonymus et al is a typo, I presume?
- Bottom of page 7: there seems to be some typo/latex error
- Maybe formalize all your assumptions into an assumption latex environment (it's OK to have this in the appendix), such that readers can refer to it


- [ 1 ] Reizinger, Patrik, Szilvia Ujváry, Anna Mészáros, Anna Kerekes, Wieland Brendel, and Ferenc Huszár. "Position: Understanding LLMs Requires More Than Statistical Generalization." In Forty-first International Conference on Machine Learning.

**Questions:**

- L138: why do you use the $L^{\infty}$ norm?

---

> ### Author Response · Authors · 2025-11-19
>
> Thank you very much for your review, and your positive comments.
>
> **Language**
>
> We take your concerns about our choice of language seriously. We believe you meant to reference [1], which defines “absolute” and “relative” identifiability of representations. We ask you to please refer to the top-level comment we just posted, which we hope will foster discussions amongst all reviewers about this key contribution of our paper and how best to address your concerns: [link](https://openreview.net/forum?id=Wa3cfE3Iay&noteId=H1ef1M1pVD).
>
> ***
>
> We respond now to your other specific notes.
>
> **Major points**
>
> *M1/3: vocabulary choice.*
>
> Please see our top-level reply linked above and let us know if the proposed change would address your concerns.
>
> *M2: clarity of Theorem 1.*
>
>  We have added a note to the intuition for Theorem 1 that clarifies the role of bi-Lispchitzness: *The bi-Lipschitzness constraint controls the degree to which $g_\theta$ deforms distances. Intuitively, the bi-Lipschitz constant $L$ is small when small changes in the latent variables result in small changes in the outputs of the network.*
>
> *M4: missing reference.*
>
> We have added a discussion of “I Predict Therefore I Am…” by Liu et al. [2] to the discussion, thank you for the reference.
>
> *M5: Table 1 clarity.*
>
> We have clarified the caption of Table 1 and made it self-contained.
>
> *M6: Theorem A.3 intuition.*
>
> We have revised the proof of Theorem A.3 and discussed its intuition much more extensively in the proof. It is indeed useful pedagogically, thank you for the feedback.
>
> **Minor points**
>
> *N1: Platonic representation connection.*
>
> We have clarified this in the main text: *Second, many existing model-specific identifiability results are actually recoverability results which each require different assumptions on the data-generating process, and therefore fail to predict the empirical alignment seen across different architectures and modalities (Huh et al., 2024): for example, the latent spaces of distinct self-supervised models can often be “stitched together” by a linear or orthogonal transformation (Fumero et al., 2024; Maiorca et al., 2023). Our theory begins to address this by utilizing assumptions that are largely data-generating process agnostic, and that are more likely to be shared across model architectures.*
>
> *N2: overloaded terminology.*
>
> Succinctly, the traditional definition of identifiability (from statistics, not the ICA literature; “relative identifiability” in the terminology of [1]) does not require recoverability (“identifiability” in some non-linear ICA literature; structural identifiability in econometrics/causality; “absolute identifiability” in the terminology of [2] being a special case). It is the particularities of the usual linear ICA model where these definitions coincide. Most of the time, you can get traditional identifiability for “cheaper” (in an assumption sense) than recoverability. Please also see the extensive discussion in the top-level comment linked above.
>
> *N3: missing reference.*
>
> We now discuss “All or None…” by Marconato et al. [3] in some detail in several sections of the paper. We have added a mention of $\epsilon$-non-identifiability to the Related Work Section, as it is indeed related.
>
> *N4: clarity of bi-Lipschitz data-generating process section.*
>
> We have added additional intuition to Section 3.3.1, including a simple figure that depicts the situation, and would appreciate your feedback.
>
> *N5/N6: typographic issues.*
>
> We have corrected the missing reference and done a thorough check for LaTeX errors.
>
> *N7: assumption environments.*
>
> We have created an assumption section for each of the three theorems in the appendix.
>
> *Q1: choice of $L^\infty$.*
>
> The choice of $L^\infty$ is due to the availability of results for near-isometries under this norm, allowing us to characterize “worst case” nearness bounds on identifiability. For many parts of the paper (particularly the definitions), another $L^p$ norm can be dropped in without compromising the idea. (We have made this clear also in the proof of Theorem A.3.)
>
> ***
>
> [1] Reizinger et al. Position: An Empirically Grounded Identifiability Theory Will Accelerate Self-Supervised Learning Research. *ICML*, 2025.
>
> [2] Liu et al. I Predict Therefore I Am: Is Next Token Prediction Enough to Learn Human-Interpretable Concepts from Data?. arXiv:2503.08980
>
> [3] Marconato et al. All or None: Identifiable Linear Properties of Next-token Predictors in Language Modeling. *AISTATS*, 2025.

---

> > ### Comment · Reviewer_p3of · 2025-11-24
> >
> > Thank you very much for your detailed response! I added a few remarks above re:identifiability. But otherwise, you have addressed all my concerns, thus, **I gladly and strongly recommend your paper for acceptance.**

---

### Official Review · Reviewer_QUzt · 2025-10-29

**Soundness:** 4
**Presentation:** 4
**Contribution:** 4
**Rating:** 8
**Confidence:** 4

**Summary:**

The authors study $\epsilon$-near identifiability and recoverability in self-supervised models. They define near-identifiability of a model as satisfying an identifiability criteria up to $\epsilon$. They then show that, for a model with identifiable output, the internal representations are near-identifiable up to rigid transformations and up to $\epsilon$ a function of the Lipschitz constants of the intermediate layers. They then show that ICA post-processing can be used to preserve near-identifiability, which results in near identifiability up to permutation and sign flips.

**Strengths:**

Existing works in identifiability usually treat functions $f$ as fully black-box, but in practice are parametrised as neural networks. Arguably the black-box perspective has been taken to its limit and there are not many substantially new developments in identifiability here. If I understand correctly, what this paper does is leverage the neural network (or compositional) structure, noticing that outputs are usually identifiable (e.g., assuming optimization is well-posed), and leveraging that structure to obtain a result on near identifiability of intermediate layers, which is how many representations are extracted in practice. This is a really neat, original idea that should have significant impact for the representation learning community.

**Weaknesses:**

- The recoverability section, unless I'm missing something, seems to just be talking about how well-specification is usually assumed in these papers, but it is nice to have it spelled out with examples specific to the theory here.

**Questions:**

- Are any of the results really specific to self-supervised models, or do they have some special qualities (e.g., in the loss)? Why is the emphasis placed on this setting in the framing?

---

> ### Author Response · Authors · 2025-11-19
>
> Thank you very much for your review and your positive comments. Indeed, you have hit on one of our key contributions: simple function class assumptions are enough to extend output identifiability to internal representation identifiability.
>
> Below, we respond point-by-point to the weakness and question you raise.
>
> *W1: “the point” of recoverability.*
>
> Yes, indeed, the goal of this section is to highlight the fact that for some key results (what we, and statisticians writ large, call identifiability) do not require well-specification of the data-generating process, only the model. On the other hand, what we call recoverability does require some well-specification of the DGP. This is important for masked autoencoders for example, because we can get identifiability without perfect reconstruction, while our recoverability result depends on it. We make this distinction explicit in part so that we can exploit it in our disentanglement experiments (Section 4.2). Please also see our top-level comment discussing some of our choices around semantics, including a proposal to slightly change some vocabulary in our paper to better reflect the history of these terms: [link](https://openreview.net/forum?id=Wa3cfE3Iay&noteId=H1ef1M1pVD).
>
> *Q1: why self-supervised models?*
>
> In general, self-supervised models are the most likely to satisfy our substitute for the “sufficient diversity” condition from [1]: in particular, the mapping between the outputs and the latents must be injective and bi-Lipschitz – intuitively, “nearby” latents ought to generate similar outputs. As such, self-supervised models which optimize the outputs to be e.g. good reconstructions of the inputs are well-suited to meeting this condition, although we emphasize that supervised learners with a sufficiently large number of classes (such as next-token predictors) are also often assumed to meet this requirement like in [1]. We have clarified this in the intuition following Theorem 1.
>
> ***
>
> [1] Roeder, G. et al. On Linear Identifiability of Learned Representations. ICML, 2021.

---

### Official Review · Reviewer_tnLY · 2025-11-01

**Soundness:** 3
**Presentation:** 2
**Contribution:** 3
**Rating:** 6
**Confidence:** 3

**Summary:**

In this work, the authors provide a novel identifiability result which characterizes the similarity in parameter space for a broad family of models up to rigid transformations in terms of a bound which is a function of the bi-Lipschitz constant of a model. The authors then show a relationship between this notion of a identifiability and the idea of achieving identifiability w.r.t. a latent variable model, which they refer to as recoverability. Several experiments are run across large scale models and various datasets highlighting that applying ICA on learned representations yields disentangled representations.

**Strengths:**

I believe this work makes contributions which are potentially of substantial interest to the identifiability community.

*  Namely, providing clarity on the the relationship between identifiability w.r.t. a latent variable model and models being identifiability w.r.t. each other is an important nuance to explore.

* Furthermore, Theorem 1 provides very general result characterizing model identifiability which I believe is of interest even if the result relies on a Lipschitz constant which may be difficult to measure in practice.

* The authors make a notable effort to conduct experiments which highlight representation similarity and disentanglement in diverse settings and across different large scale models.

**Weaknesses:**

* While I find the author's contributions noteworthy, I found it very difficult to parse the contributions by reading the abstract and introduction of the paper. In the introduction, the authors introduce several purported issues with existing identifiability results and then present their contributions as solving these issues. I found this relationship between these issues and the authors contributions difficult to connect and ultimately obfuscating of the authors main contributions. I would suggest the authors rework the abstract and intro to better clarify their contribution.


* While I understand that the authors wish to explore the implications of their results on image data, the experiments feel a bit disconnected from the theory. I believe a simple toy experiment validating the authors theory in a setting in which the bi-lipschitz constant can be measured would be of value.


* Additionally, I would appreciate if the authors could discuss the relationship between their results and prior results [1] which have results related to epsilon near identifiability.


**References**

[1] Nielsen et. al 2025, When Does Closeness in Distribution Imply Representational Similarity? An Identifiability Perspective

**Questions:**

* Can the authors clarify the positioning/motivation of their results within the identifiability community?

* Do the authors believe it would be possible to conduct experiments on toy data which aim to more rigorously test the theoretical statements?

---

> ### Author Response · Authors · 2025-11-19
>
> Thank you very much for your review and your positive comments. Below, we respond point-by-point to the weaknesses you mention.
>
> *W1: clarity of Abstract and Introduction*
>
> In response to your comment, we have taken advantage of the increased length limit and made key changes to the Abstract and Introduction. Specifically, we have highlighted how the separation of identifiability and recoverability allows us to better understand the distinction between stability and downstream utility of representations; how recoverability provides a potential pathway toward understanding cross-model convergence; and how we trade off data-generating process assumptions for model assumptions.
>
> *W2/Q1: contextualization of related work.*
>
> We have added more detail to the contextualization in the Related Work section, including to [1]. We also hope that this addresses your Q1. We welcome feedback as to whether further work is missing.
>
> *Q2: additional bi-Lipschitz experiment.*
>
> In response to your Q2, it is challenging although perhaps not impossible to manipulate the bi-Lipschitz constant in this way in a non-trivial setting, and something we are considering now (including possibly for future work if this lens of analysis for representation stability gains traction). We have for the time being opted for synthetic experiments which are more easily compared to prior work (disentanglement, Section 4.2).
>
> ***
>
> [1] Nielsen et al. When Does Closeness in Distribution Imply Representational Similarity? An Identifiability Perspective. *NeurIPS*, 2025.
>
> [2] Zhang, G et al. Three mechanisms of weight decay regularization. *ICLR*, 2019.

---

> ### Author Response · Authors · 2025-11-24
> **Additional experiment completed**
>
> We thank the reviewer again for their detailed review and in particular the suggestion to run an experiment where the local bi-Lipschitz constant is manipulated.
>
> Following the suggestion, we have run an experiment in a synthetic setting. We separate MNIST by digit class, to avoid class-level indeterminacies as seen in e.g. [1]. We then fit LeakyReLU vanilla autoencoders to the observations, with different levels of leakage $\alpha$ and weights matrices constrained to be orthogonal. This allows us to control the bi-Lipschitzness of the learned autoencoders as explicitly as possible (as opposed to indirectly with e.g. weight decay, as in current Section 4.2 – disentanglement).
>
> When the leak constant $\alpha \rightarrow 1$ we get an orthogonal network (i.e. linear) with bi-Lipschitz constant $1$, i.e. $L = 0$ for our theory. On the other hand, as $\alpha \rightarrow 0$ we recover a typical ReLU network (with decoder bi-Lipschitz constant bounded by $1/\alpha^K$, i.e. $L = 1/\alpha^K - 1$) where $K$ is the number of layers in the decoder ($K = 3$ in all our experiments, same number of layers as in the encoder).
>
> As $\alpha \rightarrow 1$, the bi-Lipschitz constant bound approaches $1$ so that $L \rightarrow 0$ (in the algebra of our Theorem 1). Our theory predicts improved identifiability. Our new empirical results show that the empirical local bi-Lipschitz constant estimated from a random subset of training points approaches 1, suggesting $L \rightarrow 0$ is indeed a valid bound assumption in this setting. **As predicted by theory, identifiability (measured by $\ell_2$ error, as in our theory) improves markedly.** The curves of best fit are proportional to $\sqrt{L}$, as predicted by our Theorem 1.
>
> These results provide evidence that our theory is predictive in a synthetic, perfectly controllable regime.
>
> We hope this addresses your main remaining concern, and we welcome any questions. We intend to add these results to the Appendix, or the main paper (space permitting). For now, we have added a skeleton of experimental details and the key figure showing the results (Appendix A.6 & Figure 2). We will update with the full results shortly.
>
> [1] Nielsen et. al 2025, When Does Closeness in Distribution Imply Representational Similarity? An Identifiability Perspective. NeurIPS, 2025.

---

> > ### Comment · Reviewer_tnLY · 2025-11-27
> >
> > Dear authors,
> >
> > Thank you very much for your detailed feedback and interesting new experiment!
> >
> > While I am generally in favor of acceptance due to the theoretical and empirical contributions, I still have some issues regarding the positioning of the paper. I feel that the authors stated technical contribution in the introduction (P3) still feels extremely disconnected from the many problems that the authors purport to address in P2.
> >
> > For example, reg. the first "gap", i.e. that identifiability fails to explain why the representations are useful: How is it exactly that the authors results on recoverability are suddenly explaining the utility of identifiable representations? Moreover, just because a representation inverts some hypothetical ground-truth generative model doesn't automatically make the representation useful for a downstream task. To give an example, when we do PCA, we are implicitly assuming that there is a linear LVM underlying our data. This model is equally valid as a "ground-truth" model, but these latents are not necessarily useful even if we identify them.
> >
> > As I understand, the author's main contribution (at a high level) is that they bridge theoretical and empirical gaps between latent variable identifiability (recoverability) and model identifiability (identifiability). I would encourage the authors to then write their introduction in a clear and concise way such that (1) the two paradigms of identifiability (identifiability and recoverability) are introduced, (2) the existing theoretical and empirical gaps between these paradigms are elaborated, and (3) the authors then clearly describe how their contribution bridges this gap.
> >
> > As it currently stands, I think that easily understanding this contribution is very challenging as P2 and P3 in the introduction lack a logical flow and feel like disconnected paragraphs.

---

> > > ### Author Response · Authors · 2025-12-03
> > > **Updated introduction**
> > >
> > > Thank you for articulating your concern clearly and with an example. We have essentially completely re-written the Introduction, focusing on our core contributions rather than potential downstream impact of our work. Given the shortened rebuttal period, we hope that this major change would have addressed the spirit of your concerns.
> > >
> > > Additionally, you may appreciate the following changes we have made in response to other reviewers and some of your other comments, and the additional space now available:
> > > * We have added the synthetic experiments described above for MNIST to the main paper.
> > > * We have added additional context regarding [1] to the “Connections to prior work” (Section 3.1) and Appendix A.3.1.
> > >
> > > Your example of PCA is a good one, and almost exactly parallels the example of linear regression which was previously mentioned in our Introduction, and is now fleshed out in Appendix A.1 (Example 1). Essentially, we argue that there remain interesting questions even in your example of PCA. For example, suppose the true data-generating process (DGP) is non-linear. When and in what sense does the learned PCA model “almost” structurally identify this DGP? (Answers to these kinds of questions take the form of structural near-identifiability, our Definition 2.) We agree that we certainly have not provided a universal definition of “utility” (and have updated our introduction to ensure that it doesn’t seem that we claim so). You may also find it interesting to read Example 2 (“mis-specification” in masked autoencoders) in Appendix A.1, which is less straightforward than the case of PCA because the model (and DGP) are non-linear.
> > >
> > > [1] Nielsen et. al 2025, When Does Closeness in Distribution Imply Representational Similarity? An Identifiability Perspective. *NeurIPS*, 2025.

---

### Official Review · Reviewer_hzYt · 2025-11-01

**Soundness:** 3
**Presentation:** 3
**Contribution:** 3
**Rating:** 6
**Confidence:** 3

**Summary:**

A quite detailed analysis around self supervised learning and representation stability. There are technical contributions that expand the applicability of identifiability theory to a broader class of models. While the experiments remain quite limited and synthetic, there is a lack of more challenging data with known ground truth hence making it a challenge for the authors.

** Strength **
- The paper introduces quite a few theoretical contributions that take a step towards better understanding self supervised learning which is one of the most prominent solution today
- The writing and technical quality is above acceptance level with clear novelty and insights
- While the scope is not as wide as it could be, the empirical section is quite detailed and feel quite reproducible

** Weakness **
- Numerous formatting and typos throughout the manuscript that makes the paper feels a bit rushed in some parts, e.g., `multiviarate`, `identfiability`, `c̃itepjumpcp` and so on
- The paper is quite dense mathematically and the appendix doesn't help much as it doesn't provide more introductory materials to help unfamiliar readers get started. While this is not a major weakness, it would be great to bridge a bit more the paper's content with the practitioners unfamiliar with those concepts
- It would also be great to better connect the theory and results with practical settings for SOTA models

**Strengths:**

Please see summary

**Weaknesses:**

Please see summary

**Questions:**

Please see summary

---

> ### Author Response · Authors · 2025-11-19
>
> Thank you very much for your review, attention to detail, and positive comments. Below, we respond point-by-point to your three weaknesses.
>
> *W1: typographic issues.*
>
> We have submitted a revised version with the typos you mentioned corrected, and have done a thorough check of typos.
>
> *W2: theory overview.*
>
> In response to your point about the technical complexity, we have added an initial appendix providing an overview of the theoretical results we prove. In response to comments from other reviewers, we have also attempted to clarify other parts of the introduction which hopefully help address your concern.
>
> *W3: experimental design.*
>
> We are motivated by an important task in cellular biology: deconfounding by disentangling batch effects in the foundation model OpenPhenom (22M parameters; trained on 2.2M images). This is our “final experiment”, motivated by more direct validation of our theory (Section 4.1) and disentanglement experiments on toy data in Section 4.2.
>
> Our hope is that we have sufficiently addressed your concerns for the score to be raised. If not, we kindly ask you to clarify whether there is any further detail we could add that would strengthen the paper sufficiently for the score to be raised.
>
> Thank you for your time.

---

### Author Response · Authors · 2025-11-19
**Semantics of identifiability: proposed terminology change (I)**

We thank *Reviewer p3of* for raising important questions about our choice of language around “identifiability” and “recoverability”, which we agree with. We raise this as a general comment as we propose a change of terminology in the paper.

First, we briefly describe the core of the proposed change. Next, in case the reviewers are interested, we follow up with a deepdive on the semantic history of the term “identifiability” from the statistical, causal, and ML perspective, highlighting also with examples why the definition of [4] is insufficient for ML purposes.

**Proposed terminology change**

Identifiability is clearly an overloaded term. However, we could replace the term *recoverability* that *Reviewer p3of* did not appreciate with the term “structural identifiability” (alongside our “near-” relaxation). This terminology first appeared in econometrics [2] and we feel it sufficiently distinguishes it from the statistical and ML notions of parameter and representation identifiability, respectively. While this will result in many small changes in the paper around Definition 2/Theorem 3, they are only superficial semantic changes. Thus, we ask for feedback from all reviewers about this option.

In any case, we will add an Appendix that discusses the history of the semantics and reference it throughout the paper where relevant. We will contextualize past results from nonlinear ICA in this Appendix. Specifically, in the next draft we submit within the next week, we will:

1. Show the relationship between absolute identifiability, relative identifiability, and recoverability from [4] with our definitions and the examples below showing that the prior definition in [4] is insufficient.
2. Contextualize some of the non-linear ICA results under these definitions as examples.

***

**References**

[1] Casella, G. & Berger, R. Statistical Inference (1990).

[2] Koopmans, T. C. & Reiersøl, O. The Identification of Structural Characteristics. *The Annals of Mathematical Statistics, 1950.

[3] Pearl, J. Causal Diagrams for Empirical Research. *Biometrika*, 1995.

[4] Reizinger, P. et al. Position: An Empirically Grounded Identifiability Theory Will Accelerate Self-Supervised Learning Research. *ICML*, 2025.

[5] Severini, T. A. Some properties of inferences in misspecified linear models. *Stat Probab Lett*, 1998.

[6] Comon, P. Independent Component Analysis, a new concept? *Signal Processing*, 1994.

[7] Roeder, G. et al. On Linear Identifiability of Learned Representations. *ICML*, 2021.

[8] Reizinger, P. et al. Position: Understanding LLMs Requires More Than Statistical Generalization. *ICML*, 2024.

---

> ### Author Response · Authors · 2025-11-19
> **Semantics of identifiability (II)**
>
> **History of the term identifiability**
>
> Identifiability as a concept has a long history in statistics before work on independent components analysis. For example, consider the following definition from Casella & Berger’s Statistical Inference (1. ed released 1990; a canonical stats textbook with 17K citations) [1, p. 548]:
>
> > A parameter $\theta$ for a family of distributions $\\{ f(x \mid \theta) : \theta \in \Theta \\}$ is identifiable if distinct values of $\theta$ correspond to distinct pdfs or pmfs. That is, if $\theta \neq \theta’$, then $f(x \mid \theta)$ is not the same function as $f(x \mid \theta’)$.
>
> Our paper’s definition of identifiability (Definition 1) makes a straightforward generalization from likelihoods $f$ to losses $\mathcal{L}$. Further, because of the dominance of empirical risk minimization and the fact that, unlike most likelihoods in statistical inference, losses $\mathcal{L}$ are non-convex, we define identifiability at the minima of the expected loss. In other words, our definition of identifiability agrees with the statistical one, modulo some small changes to make it useful for talking about non-convex optimization. Indeed, this does not only agree with Roeder et al. (2020) [7], but also Reizinger et al. (2024) [8]: *“In machine learning, identifiability can be interpreted as a guarantee that the test loss has a unique minimizer.”*
>
> On the other hand, there is a similarly long history of “structural identifiability” from econometrics. For example, consider Koopmans & Reiersøl (1950; Nobel-winning) [2]:
>
> > _Identifiability of structural characteristics by a model._ It is therefore a question of great practical importance whether a statement converse to the one just made is valid: can the distribution $H$ of apparent variables, generated by a given structure $S$ contained in a model $\mathcal{S}$, be generated by only one structure in that model?
>
> Clearly, this is a different definition. Indeed, Koopmans & Reiersøl acknowledge that it is different from the statistical definition above, which persisted from the work of Fisher in the 1920s. It implicitly assumes the existence of a “true model” living within the estimable model class (and that this model generated the data), which flies in the face of mathematical statistics’ common quip “all models are wrong”. Pearl (1995; cited in his Turing award’s annotated bibliography) [3] inherits this definition, perhaps contributing to its later infusion into the ICA literature:
>
> > *DEFINITION 4 (Identifiability).* The causal effect of X on Y is said to be identifiable if the quantity $P(y \mid x)$ can be computed uniquely from any positive distribution of the observed variables that is compatible with a graph G.
>
> Put simply, both these definitions of structural identifiability *assume that the true data-generating process matches the model*, and under this assumption, consider whether some structural parameter of interest can be identified.
>
> Interestingly, the main result of the original seminal linear ICA paper (Comon et al., 1994) [6] is a statistical identifiability result, not structural (we have pruned some language from the paper’s Corollary 13 for readability):
>
> > Let no noise be present, and define $y = M x$ and $y = Fz$, $y$ being the observed random variable satisfying some requirements. Then if $T$ is discriminant, $T(p_z) = T(p_x)$ if and only if $F = M A P$, where $A$ is an invertible diagonal matrix and $P$ a permutation.
>
> It would seem that because the assumptions for statistical identifiability in the linear ICA setting also coincide with those necessary for structural identifiability, the two terms have historically been used somewhat interchangeably in this setting as the models have become more complex (i.e. non-linear). However, as we will see in the examples below, this is not necessarily the case for all models.
>
> **The key issues around the term "identifiability"**
>
> First: it is clear that statistical and structural identifiability are different concepts historically. We agree that there is a subset of statistical learning papers (particularly in nonlinear ICA) that use the term non-specifically. Yet, these concepts exist independently, and it is somewhat just “bad luck” that in this setting, we sometimes find the need to talk about both.
>
> Second: it is clear that these concepts need to remain distinct for talking about representation identifiability in machine learning. This is made clear also in e.g. [4].
>
> Third: definitions like in [4] are unduly restrictive for talking about mis-specified models. In particular, there are cases where a model’s representations are identifiable according to our definition (statistically identifiable), recoverable according to our definition, but not absolutely identifiable according to the definition of [4]. In short, these situations arise where the model is in some sense “mis-specified”.

---

> ### Author Response · Authors · 2025-11-19
> **Semantics of identifiability (III)**
>
> So, even if we adopt the *terminology* from [4], we need the more formal definitions proposed in our paper (and not only for our "near-" relaxation). The following examples make this concrete.
>
> **Example 1: mis-specified linear regression**
>
> The first example comes from traditional statistics. Consider the identifiability of the usual linear model $Y = \beta X + \epsilon$ when the model is mis-specified in the sense that the true data-generating process is $Y = f(X) + \epsilon$ for some non-linear $f$. If $f$ meets certain conditions, the following facts are true:
>
> 1. $\beta$ is statistically identifiable under the usual OLS assumptions (i.e. identifiable according to our definition),
> 2. $f$ is not recoverable according to our definition,
> 3. the model is not absolutely identifiable according to [4], because the DGP does not match the model,
> 4. a function $g$ *is* recoverable according to our definition under assumptions on $f$, where $g$ is in some sense the “nearest” linear function to $f$ [5].
>
> **Example 2: imperfect reconstruction in MAEs**
>
> The second example is drawn from our paper, although we do not spell it out explicitly (yet). For a masked autoencoder, the following facts could be true all at once, such as in the case of imperfect reconstruction (we drop the “near-“ qualifier, because it’s not of importance):
>
> 1. the internal representations given by the encoder $f$ are identifiable according to our definition,
> 2. the true data-generating process $g$ is not recoverable because of imperfect reconstruction, but
> 3. some approximation to the data-generating process $h$ *is* recoverable, but $h \neq g$ because masked training prevents perfect reconstruction, so
> 4. the model is not absolutely identifiable according to [4].
>
> In this case, we might not have the simple analytical form for $h$ that we do in Example 1, but it’s clear that *something* about the DGP is recoverable (we do not treat this case, because it requires making assumptions on how the “imperfections” in the reconstructions arise, i.e. defining some notion of closeness between $h$ and $g$, which is beyond the scope of this work -- it is different from our concept of nearness, to be clear).

---

> > ### Comment · Reviewer_p3of · 2025-11-24
> >
> > First and foremost, thank you for your **extremely enlightening and nuanced response**. I haven't been familiar with some of the cited works (espcially the ones in econometrics). I am sure that **this discussion and background would be extremely valuable** for the community.
> >
> > A few remarks and questions:
> > 1. I like you contrasting the different definitions, though I cannot fully graps everything from the cited formulations (I know that it's unreasonable to quote all details here, so it's only an observation, not a critique). **For the paper, please reformulate these definitions in consistent language and all context.**
> > 2. I think **structural identifiability** captures the (perhaps implicit) core of what people mean by identifiability in ICA and also CRL. Indeed, previous works have shown connections between the two-starting with the classical result by Shimizu et al., 2006 that linear ICA can be used to learn the causal structure. See also [Reizinger et al., 2023](https://openreview.net/forum?id=2Yo9xqR6Ab) and [the overview](https://openreview.net/forum?id=k03mB41vyM)
> >
> > Please incorporate these points into your paper. I can already say that your rebuttal improved your submission substantially, which I will reflect in my raised score.

---

> ### Author Response · Authors · 2025-12-03
>
> Thank you for the kind words! We updated our submission substantially to reflect this discussion, including changing the term "recoverability" to "structural identifiability", distilling the above deep dive on the history of the term "identifiability" into Appendix A.1, including discussion on results from the ICA literature, and ensuring vocabulary is consistent throughout the paper.
>
> Thank you very much for your engagement during the discussion period, particularly given the unusual circumstances.

---

### Author Response · Authors · 2025-12-03
**Rebuttal period summary**

We sincerely appreciate the feedback from reviewers on our submission, particularly from *Reviewer p3of* and *Reviewer tnLY* who took the time to interact with us during the discussion period. Thanks to feedback from all of the reviewers and these discussions, we feel our submission is much improved.

To assist the AC with their assessment, we provide the following brief summary of the rebuttal period:

1. **Terminology.** *Reviewer p3of* raised an important question: was it necessary to propose a new term, “recoverability”, in our initial submission? As a result, we did a deep dive on the history of the term “identifiability” and had a [discussion](https://openreview.net/forum?id=Wa3cfE3Iay&noteId=H1ef1M1pVD) with the reviewer. As a result of this discussion, we changed the term “recoverability” to “structural identifiability”, in recognition of a long history of use in econometrics and other disciplines, including recent developments in ICA. This resulted in several small changes throughout the paper, most notably the updated title (as visible in the PDF) and in Definition 2 (line 248). Additionally, we distilled the key elements of our discussion on terminology into a new Appendix in the paper (A.1). This change also has a consequence of better aligning our terminology with the ICA community. Our feeling is that this broadens the potential impact of our paper. Furthermore, the change was very well-received by the reviewer, who updated their position strongly in favor of acceptance and score to 10, reinforcing the comment in their original review that the “paper has the potential to be a spotlight/oral”.
2. **Additional synthetic experiment.** *Reviewer tnLY* pointed out that direct experimental validation of Theorem 1 would be useful, showing how the bi-Lipschitzness of the model impacts the level of near-identifiability $\epsilon$. As a result, we conducted a new experiment on MNIST showing exactly that, which was well-received by the reviewer. This updated experiment is presented in Section 4.1 (Lines 351-369) and Appendix A.7.1.
3. **Clarity of the abstract and introduction.** *Reviewer tnLY* kindly pointed out that our Introduction did not communicate the contributions of our paper clearly. As a result, we almost completely re-wrote the introduction from scratch (Lines 47-79), and updated our Abstract (main changes on Lines 15-17). Although this was unfortunately just before the rebuttal period closed early, and so we do not have feedback on our changes from this reviewer, we shifted substantial focus in the Introduction away from the potential impact of our results to a clear summary of our contributions, which we feel satisfies the spirit of the reviewers request.

We were pleased that the initial scores by most reviewers were positive, and the only reviewer with a negative score responded extremely favorably to our discussion and updates as discussed in Point 1 above (Terminology), raising their score to 10, reinforcing the comment in their original review that the “paper has the potential to be a spotlight/oral”. Again, we thank the reviewers and AC for their service under these unusual circumstances.

---

### Meta-Review · Area_Chair_b1Ud · 2025-12-31

**Summary:**

All reviewers appreciated the technical contributions of the paper, which are summarized below.

**Summary of Technical Contributions** \
Conceptual Definitions
1. A clear distinction between two notions of identifiability in representation learning:
    * Statistical identifiability, referring to agreement between representations learned by different models.
    * Structural identifiability, referring to agreement between learned representations and the ground-truth generative process.
2. A generalization of both notions to their approximate counterparts.

Technical Results
1. Theorem 1 (Stability of latent representations in autoencoders).  Statistical identifiability between the outputs of two autoencoders implies approximate statistical identifiability between their latent representations. The approximation error is governed by the smoothness of the decoder function.
2. Theorem 2 (Resolving linear ambiguity).  Given two representations that are approximately statistically identifiable up to a linear transformation, applying Independent Component Analysis (ICA) yields approximate statistical identifiability with the linear ambiguity resolved.
3. Theorem 3 (Approximate structural identifiability).  An extension of Theorem 2 to the case of structural identifiability.


**Summary of the Main Unaddressed Concern** \
The current version of the paper has one major unresolved issue. The title, introduction, and conclusions place strong emphasis on self-supervised learning. However, self-supervised learning is a broad paradigm that includes a wide range of models, such as autoencoders, contrastive learning methods, decorrelation-based approaches, and clustering-based methods. Currently, the paper provides theoretical and empirical support only for autoencoders. As a result, the current claims are overly general and not sufficiently supported by the presented experiments.

**Motivation of Recommendation** \
The paper makes a valuable contribution to the theory of identifiability in representation learning. The authors have made several changes to the text during the rebuttal phase aimed mainly at enhancing its clarity. However, to resolve the remaining issue outlined above, **it is strongly recommended that the authors revise the title and corresponding parts of the paper by replacing the term *self-supervised learning* with either *autoencoders* or the term *representation learning***.

The paper can be accepted conditionally upon the implementation of these requested changes.

**Reviewer Concerns:**

Reviewers have raised the following main concerns:
1. **Clarity**. The original version of the paper introduced new terminology and lacked sufficient background, which obscured the paper’s contributions and hindered readability (*All Reviewers*). Authors have made changes to primarily address this concern. In particular, they introduced a revised terminology, replacing terms such as identifiability and recoverability with statistical identifiability and structural identifiability. This change helps reduce confusion arising from the multiple definitions used across statistics, causality, and unsupervised learning, as summarized in the historical overview in Appendix A.1. In addition, the abstract and introduction were revised to reflect this terminology and to more clearly articulate the paper’s main contributions.
2. **Clarity / Quality**. A disconnect between the theoretical results and the experimental section was noted (*Reviewers hzYt, tnLY*). This issue has been addressed by introducing a new toy experiment on MNIST in which the approximation error can be computed analytically. This addition provides a smoother transition from theory to experiments and strengthens both the validation and interpretability of the theoretical results.

**Reviewer Scores:**

*Reviewer hzYt* would have kept his score (6) because of lack of experimental support with self-supervised learning models.

*Reviewer tnLY* would have kept his score (6) but increased his confidence as the clarity of the paper has been enhanced during the rebuttal.

*Reviewer QUzt* would have kept his high score (8) and high confidence (4).

*Reviewer p3of* was initially leaning toward rejection due to a lack of clarity and insufficient positioning of the paper with respect to the existing literature. Following the rebuttal, the reviewer expressed enthusiasm for the paper and indicated that their score could have been raised likely to a level comparable to that of reviewer QUzt (8).

---

### Decision · Program_Chairs · 2026-01-26

Accept (Poster)